

**Tropospheric aerosol hygroscopicity measurements in China**
Chao Peng,[1] Yu Wang,[2] Zhijun Wu,[2] Lanxiadi Chen,[1,6] Ru-Jin Huang,[3] Weigang Wang,[4] Zhe
Wang,[5] Weiwei Hu,[1] Guohua Zhang,[1] Maofa Ge,[4,6,7] Min Hu,[2] Xinming Wang,[1,6,7] Mingjin
Tang,[1,6,7,*]
1 State Key Laboratory of Organic Geochemistry, Guangdong Key Laboratory of Environmental Protection and

7        Resources Utilization, and Guangdong-Hong Kong-Macao Joint Laboratory for Environmental Pollution and

8        Control, Guangzhou Institute of Geochemistry, Chinese Academy of Sciences, Guangzhou 510640, China

[2] State Key Joint Laboratory of Environmental Simulation and Pollution Control, College of Environmental

10       Sciences and Engineering, Peking University, Beijing 100871, China

[3] State Key Laboratory of Loess and Quaternary Geology, Center for Excellence in Quaternary Science and

12       Global Change, and Key Laboratory of Aerosol Chemistry and Physics, Institute of Earth Environment,

13       Chinese Academy of Sciences, 710061 Xi'an, China

[4] State Key Laboratory for Structural Chemistry of Unstable and Stable Species, Beijing National Laboratory

15       for Molecular Sciences (BNLMS), CAS Research/Education Center for Excellence in Molecular Sciences,

16       Institute of Chemistry, Chinese Academy of Sciences, Beijing 100190, China

[5] Division of Environment and Sustainability, The Hong Kong University of Science and Technology, Hong

18       Kong, China

[6] University of Chinese Academy of Sciences, Beijing 100049, China
[7] Center for Excellence in Regional Atmospheric Environment, Institute of Urban Environment, Chinese

21       Academy of Sciences, Xiamen 361021, China

* Correspondence: Mingjin Tang (mingjintang@gig.ac.cn)





**Abstract**

Hygroscopicity largely determines phase state, chemical reactivity, optical properties and

cloud nucleation activities of aerosol particles, thus significantly affecting their impacts on
visibility, atmospheric chemistry and climate. In the last twenty years a large number of field
studies have investigated hygroscopicity of tropospheric aerosols in China under sub- and super-
saturated conditions. Aerosol hygroscopicity measurements in China are reviewed in this paper: 1)
a comprehensive summary and critical discussion of aerosol hygroscopicity measurements in
China is provided; 2) available measurement data are compiled and presented under a consistent
framework to enhance their accessibility and usability; 3) current knowledge gaps are identified,
and an outlook which could serve as guidelines for planning future research is also proposed.



## 1 Introduction

In the last few decades, rapid industrial, economic and social developments in China have caused large emissions of gaseous and particulate pollutants into the troposphere (Li et al., 2017a), where they are mixed with gases and aerosols from natural sources. Under unfavourable meteorological conditions (i.e. when air is stagnant and stable), severe air pollution occurs, due to accumulation of primary pollutants and more importantly, formation of secondary pollutants (Zhu et al., 2011; He et al., 2014; Zhang et al., 2015; An et al., 2019; Lu et al., 2019; Zhang et al., 2019c). During severe air pollution events, $PM_{2.5}$ could exceed a few hundred μg m$^{-3}$ (Guo et al., 2014; Huang et al., 2014) and $O_3$ could reach up to >200 ppbv (Wang et al., 2017a). The concept of air pollution complex has been proposed to describe the complexity of air pollution in China, characterized by complex sources and complex interactions of a myriad of gaseous and particulate pollutants (Zhu et al., 2011; Lu et al., 2019; Chu et al., 2020). Thanks to the implementation of effective air pollution control measures, substantial decrease in $PM_{2.5}$ has occurred nationwide in the last several years (Zhang et al., 2019b); however, slight but significant increase in $O_3$ has been observed in many regions during the same period (Li et al., 2019a; Lu et al., 2020), revealing the complexity and difficulty in synergistic control of $PM_{2.5}$ and $O_3$.

Hygroscopicity, one of the most important physicochemical properties of aerosols, determines the amount of water associated with aerosol particles under ambient conditions (mainly relative humidity, and temperature to a less extent) and significantly affects their environmental and climatic impacts (Kreidenweis and Asa-Awuku, 2014; Tang et al., 2019). Hygroscopicity is referred to hygroscopic properties under subsaturated conditions from a specific view, while from a general view, it is referred to both hygroscopic properties under subsaturated conditions and cloud condensation nucleation (CCN) activities under supersaturated conditions. Due to their



hygroscopicity, aerosol particles will take up water (i.e. hygroscopic growth) and lead to increase
in particle mass and size (Kreidenweis and Asa-Awuku, 2014; Tang et al., 2016; Wu et al., 2018b;
Tang et al., 2019). Therefore, hygroscopicity largely determines optical properties of aerosols and
as a result their impacts on visibility and direct radiative forcing under subsaturated conditions
(Titos et al., 2016; Zhao et al., 2019); on the other hand, hygroscopicity is also closely linked to
CCN activities of aerosols and thus their abilities to from cloud droplets under supersaturated
conditions (Kreidenweis and Asa-Awuku, 2014; Farmer et al., 2015; Tang et al., 2016), thereby
having important implications for their indirect radiative forcing (Dusek et al., 2006; McFiggans
et al., 2006; Farmer et al., 2015). Furthermore, hygroscopicity determines aerosol liquid water
content (ALWC) and thus phase state, acidity and chemical reactivities of aerosols (Bertram and
Thornton, 2009; Liu et al., 2017; Tang et al., 2017; Wu et al., 2018b), playing critical roles in
secondary aerosol formation as well as removal and production of trace gases. In addition,
hygroscopic growth measurements can provide valuable insights into mixing states of aerosols
(Swietlicki et al., 2008; Riemer et al., 2019). Due to its importance, tropospheric aerosol
hygroscopicity has been investigated in China by a number of field studies in the last 10-20 years,
as reviewed in this paper.

Swietlicki et al. (Swietlicki et al., 2008) summarized and analyzed hygroscopic properties of

ambient aerosols measured using H-TDMA (Hygroscopic Tandem Differential Mobility Analyser)
prior to September 2007, when ambient aerosol hygroscopicity was seldom explored in China. The
effects of hygroscopicity on aerosol light scattering have been reviewed and summarized on the
global scale (Titos et al., 2016; Burgos et al., 2019), and a very recent paper also briefly
summarizes aerosol light scattering enhancement studies in China (Zhao et al., 2019). A book
chapter (Kreidenweis and Asa-Awuku, 2014) discussed in brief hygroscopic growth and light



scattering enhancement of ambient aerosols, but only a few measurements conducted in China
were included. In addition, a recent paper (Tang et al., 2019) has reviewed aerosol hygroscopicity
measurement techniques. However, aerosol hygroscopicity measurements in China have not been
reviewed yet.
In this paper we provide a comprehensive review of hygroscopic properties of ambient aerosols
measured using H-TDMA in China; in addition, CCN activities of tropospheric aerosols measured
in China are also reviewed and discussed. Via using the single hygroscopicity parameter ($\kappa$), we
attempt to reconcile hygroscopic properties examined at <100% RH (relative humidity) with CCN
activities measured at >100% RH. A number of studies measured light scattering enhancement
factors, $f$(RH), of ambient aerosols in China (Zhao et al., 2019), but most of these studies are not
included herein for two reasons: 1) $f$(RH) measurements in China have been reviewed in brief very
recently (Zhao et al., 2019); 2) it is not trivial to convert measured $f$(RH) to growth factors or $\kappa$
values (Kreidenweis and Asa-Awuku, 2014). Nevertheless, we note that some methods have been
proposed to convert measured $f$(RH) to $\kappa$ (Kuang et al., 2017; Kuang et al., 2018). Single particle
techniques were employed to investigate hygroscopic properties of tropospheric aerosols (Li et al.,
2016); however, as numbers of particles examined in single particle studies are usually too limited
to provide enough information for the overall aerosol hygroscopicity, these studies are not
discussed herein. Although not covered in this review, remote sensing techniques can also be used
to retrieve aerosol hygroscopicity in the troposphere (Lv et al., 2017; Bedoya-Velásquez et al.,
2018; Tang et al., 2019; Dawson et al., 2020).
The first goal of this paper is to provide a comprehensive overview of hygroscopic properties
and CCN activities of tropospheric aerosols in China via reviewing previous field studies. The
second goal is to compile and present measurement data (as compiled in Tables S1-S5) reported





by previous work using a consistent framework (i.e. via using the single hygroscopicity parameter)
to enhance their accessibility and usability. The third goal, perhaps more importantly, is to identify
knowledge gaps in this field and then to provide an outlook which can serve as practical guidelines
for planning future research. In this paper, Section 2 describes the methodology adopted in this
paper to analyse and review previous studies, and previous measurements of hygroscopic
properties and CCN activities of tropospheric aerosols in China are reviewed and discussed in
Sections 3 and 4. In the end, Section 5 outlines knowledge gaps and research perspectives.

## 2  Methodology

### 2.1  Hygroscopic properties

H-TDMA instruments, initially developed ~40 years ago (Liu et al., 1978; McMurry et al.,

1983; Rader and McMurry, 1986; McMurry and Stolzenburg, 1989), have been widely used in
field and laboratory studies (Kreidenweis et al., 2005; Svenningsson et al., 2006; Gysel et al., 2007;
Sjogren et al., 2008; Swietlicki et al., 2008; Duplissy et al., 2009; Asmi et al., 2010; Liu et al.,
2011; Wu et al., 2011; Kreidenweis and Asa-Awuku, 2014; Zieger et al., 2017; Tang et al., 2019).
Technical details of H-TMDA measurements, including operation principles, data analysis and etc.,
have been detailed in a review paper (Swietlicki et al., 2008). In brief, an aerosol flow, dried to
<20% RH, is passed through an aerosol neutralizer and the first DMA (Differential Mobility
Analyzer) to produce quasi-monodisperse aerosols with a given mobility diameter; after that, the
aerosol flow is delivered through a humidifier to be humidified to a given RH, and subsequently
aerosol size distributions are measured using the second DMA coupled with a CPC (Condensation
Particle Counter). The hygroscopic growth factor, GF, is defined as the ratio of the aerosol mobility
diameter at a given RH to that at dry conditions. As aerosol particles at a given size may have
different hygroscopic properties and thus display different GF values at a given RH, probability





distribution functions of GF (i.e. number fractions of aerosol particles at each GF) have also been
reported in some studies.

The measured distribution functions of GF are usually smoothed and skewed due to several

reasons, e.g., the finite width of the DMA's transfer function, and several TDMA inversion
algorithms have been proposed to convert the H-TDMA raw data to the probability density
function of GF (Stolzenburg and McMurry, 1988; Stratmann et al., 1997; Voutilainen et al., 2000;
Cocker et al., 2001; Cubison et al., 2005; Gysel et al., 2009). The algorithm developed by Gysel
et al., TDMAinv, is currently the most widely used one. Errors and uncertainties of H-TDMA data
can come from several sources, including RH and temperature variability, electrical mobility
classification, particle non-equilibrium in the second DMA, and etc. Swietlicki et al. (Swietlicki
et al., 2008) comprehensively discussed the sources and magnitudes of these errors and how they
can be reduced or minimized. In addition, guidelines used for H-TDMA measurements, including
instrumental design, calibration, validation and operation as well as data analysis, have been
recommended in literature (Duplissy et al., 2009; Massling et al., 2011).

H-TDMA measurements of ambient aerosols were typically conducted for a few different

particles diameters at a given relative humidity (RH); most measurements were carried out at 90%
RH, though some studies also reported growth factors (GF) at other RH. To facilitate comparison
of GF reported at different RH, we convert GF measured at a given RH to $\kappa$ using Eqs. (1-2)
(Petters and Kreidenweis, 2007; Tang et al., 2016):
$\kappa = (\text{GF}^3 - 1)(\frac{B}{RH} - 1)$        (1)
$B = \exp\left(\frac{A}{d_0 \cdot GF}\right)$        (2)



where $d_0$ is the dry particle diameter; $A$, which describes the Kelvin effects, is equal to 2.1 nm at
298.15 K if the surface tension is assumed to be the same as water (0.072 J m$^{-2}$) (Petters and
Kreidenweis, 2007; Tang et al., 2016). Converting GF to $\kappa$ also facilitates comparison between
hygroscopic properties and CCN activities. For a few studies which reported GF at different RH,
we focus GF measured at 90% RH; if the data at 90% are not available, we then choose
measurements at the RH closest to 90%.

To further facilitate comparison between different measurements, Swietlicki et al. (Swietlicki

et al., 2008) classified aerosol hygroscopicity into four groups according to their GF at 90% RH.
This methodology was adopted by Ye et al. (Ye et al., 2013) who reported aerosol hygroscopic
growth measurements in Shanghai. Nevertheless, Ye et al. (Ye et al., 2013) classified aerosol
particles into three modes (instead of four), and the criterions used are slightly different from
Swietlicki et al. (Swietlicki et al., 2008). Here we adopt the method proposed by Ye et al. (Ye et
al., 2013), who classified aerosol hygroscopicity into three modes, including the nearly-
hydrophobic (NH, $\kappa$<0.1), the less-hygroscopic (LH, 0.1< $\kappa$<0.0.25) and the more-hygroscopic
(MH, $\kappa$>0.0.25) modes. However, here a few further statements are necessary. First, terminologies
used differ in previous studies for aerosol hygroscopicity modes. For example, bimodal aerosol
hygroscopicity was frequently observed in China (as discussed in Section 3), and the nearly-
hydrophobic mode defined by Ye et al. (Ye et al., 2013) was called the less-hygroscopic mode or
the low-hygroscopic mode in several studies. Second, actual aerosol hygroscopicity in the
troposphere may not perfectly fit into one of the three modes defined by Ye et al. (Ye et al., 2013).
**2.2  CCN activities**

A variety of instruments have been developed to measure CCN number concentrations

(Twomey, 1963; Sinnarwalla and Alofs, 1973; Fukuta and Saxena, 1979; Hudson, 1989; Ji et al.,



1998; Chuang et al., 2000; McMurry, 2000; Nenes et al., 2001; Otto et al., 2002; VanReken et al.,
2004; Roberts and Nenes, 2005; Frank et al., 2007; Kreidenweis and Asa-Awuku, 2014). Currently
the most widely used one is the continuous-flow streamwise thermal gradient CCN counter based
on the design of Roberts and Nenes (Roberts and Nenes, 2005; Lance et al., 2006) and
commercialized by Droplet Measurement Technologies, and mode details of this instrument can
be found elsewhere (Roberts and Nenes, 2005; Lance et al., 2006).

Measurements of size-resolved CCN activities have been discussed in a number of previous

studies (Lance et al., 2006; Frank et al., 2007; Petters et al., 2007; Rose et al., 2008; Good et al.,
2010; Moore et al., 2010; Rose et al., 2010; Bougiatioti et al., 2011). In many studies, an aerosol
flow sampled from the ambient air, after dried to <20% RH, is passed through an aerosol
neutralizer and then a DMA to produce quasi-monodisperse aerosols. The aerosol flow is
subsequently split into two flows; one flow is sampled into a CCN counter to measure number
concentrations of cloud condensation nuclei ([CCN]), and the other one is sampled into a CPC to
measure number concentrations of condensation nuclei ([CN]). At a given supersaturation,
activation fractions ([CCN]/[CN]) are measured as a function of particle diameter (selected using
the DMA) and then fitted by an activation curve to determine the activation diameter at which the
activation fraction is equal to 0.5 (Snider et al., 2006; Rose et al., 2008; Sullivan et al., 2009;
Bougiatioti et al., 2011; Cerully et al., 2011), and activation fractions can be measured at one or
more supersaturation as a function of particle diameter. Methods used for instrument calibration
and data correction, which can be found in literature (Frank et al., 2007; Petters et al., 2007; Rose
et al., 2008; King et al., 2009; Petters et al., 2009; Moore et al., 2010), are not discussed herein.
Furthermore, $\kappa$ can be derived from the determined activation diameter at a given supersaturation
(Petters and Kreidenweis, 2007).



Maximum activation fractions may not approach one for ambient aerosols, and generally two
methods have been used to fit the data. If the maximum activation fraction of the fitted activation
curve is not fixed (three-parameter fit), the derived activation diameter ($d_a$) and single
hygroscopicity parameter ($\kappa_a$) describe the average properties of activated particles; if it is forced
to be 1 (two-parameter fit), the derived activation diameter ($d_t$) and single hygroscopicity
parameter ($\kappa_t$) describe the overall aerosol properties (Rose et al., 2010). For aerosols with bimodal
hygroscopicity distribution, $\kappa_a$ is comparable to the $\kappa$ determined using H-TDMA for the more-
hygroscopic mode, while $\kappa_t$ is comparable to the average $\kappa$ for the two modes. In addition to $d_a$ and
$d_t$, the apparent cut-off diameter (above which [CN] is equal to [CCN] at a given supersaturation.),
$d_{cut}$ (and thus $\kappa_{cut}$), can be determined if it is assumed that particles at each size are internally mixed
and that larger particles are activated first (Rose et al., 2010; Hung et al., 2014). The determination
of $d_{cut}$ does not required size-resolved activation fractions, but needs the overall activation fractions
and aerosol number size distribution (Burkart et al., 2011; Hung et al., 2014). Our review paper is
focused on $\kappa_a$ and to a less extent $\kappa_t$, and only discusses $\kappa_{cut}$ when neither $\kappa_a$ nor $\kappa_t$ was reported.
In addition, [CCN] and [CCN]/[CN] were also measured at one or more supersaturation in
Tianjin (Deng et al., 2011; Yang et al., 2012; Zhang et al., 2012), Zhangjiakou (Hebei) (Lu and
Guo, 2012), Shijiazhuang (Hebei) (Lu and Guo, 2012), Xingtai (Hebei) (Wang et al., 2018b),
Qingdao (Li et al., 2015a), Shanghai (Leng et al., 2013; Leng et al., 2014), Guangzhou (Duan et
al., 2017; Duan et al., 2018) and Mt. Huang (Fang et al., 2016), as well as over marginal seas of
China (Zhu et al., 2019; Gao et al., 2020) and northwestern Pacific (Wang et al., 2019a; Zhu et al.,
2019). As these studies did not carry out size-resolved measurements and thus did not report
critical diameters or $\kappa$, they are not further discussed herein.





## 3 Hygroscopic growth

A number of aerosol hygroscopic growth measurements have been carried out in China since 2001 using H-TDMA (or very similar instruments). Most of these measurements were performed in three regions with severe air pollution, including the North China Plain (NCP), Yangtze River Delta (YRD) and Pearl River Delta (PRD), and these studies are discussed in Sections 3.1-3.3. In addition, as discussed in Section 3.4, several measurements were also conducted at other locations in the east or south China.

### 3.1 North China plain (NCP)

The North China Plain is a heavily polluted region where many aerosol hygroscopic growth measurements were conducted, and as summarized in Table S1. In this section we review the measurements carried out at urban sites in Beijing (Section 3.1.1), rural sites in Beijing (Section 3.1.2), other urban/suburban sites (Section 3.1.3) and other rural sites (Section 3.1.4).

#### 3.1.1 Urban sites in Beijing

Aerosol hygroscopic growth has been measured at three urban sites in Beijing, including the PKU site, the IAP site, and the CAMS site.

**PKU site:** The PKU site is located on the campus of Peking University (39°59'20"N, 116°18'26"E), which is between the fourth and fifth ring road in the northwest of Beijing. All the measurements (Massling et al., 2009; Meier et al., 2009; Wu et al., 2016; Wu et al., 2017; Wang et al., 2018c) took place on the roof of a six-floor building (~30 m above ground), which is ~100 m away from a major road.

Aerosol hygroscopic growth was first measured at the PKU site during 2004-2005 (Massling et al., 2009; Meier et al., 2009). Massling et al. (Massling et al., 2009) measured aerosol hygroscopic growth (at 90% RH) in June-July 2004 and January-February 2005. Aerosol



hygroscopicity exhibited trimodal distribution, and $\kappa$ were found to be in the range of 0-0.028,
0.036-0.176 and 0.175-0.386 for the low-, medium- and high-hygroscopic modes (Massling et al.,
2009). In addition, no obvious difference in aerosol hygroscopicity was found between summer
and winter. Ammonium sulfate was the major inorganic species for the high-hygroscopic mode,
while fresh carbonaceous materials (e.g., soot) dominated the low-hygroscopic mode (Massling et
al., 2009). Aerosol hygroscopicity was found to increase with particle size and pollution levels
(Massling et al., 2009), as more secondary inorganic species were formed.
Meier et al. (Meier et al., 2009) further explore aerosol hygroscopic growth (at 90% RH) at
the PKU site in January 2005. Similar to the work by Massling et al. (Massling et al., 2009), three
aerosol hygroscopicity modes were identified, with the $\kappa$ values being 0-0.027, 0.036-0.154 and
0.152-0.366 for low-, medium- and high-hygroscopic modes (Meier et al., 2009); however, no
obvious dependence of aerosol hygroscopicity on air pollution levels was found. The average $\kappa$
were found to first increase (30-80 nm) and then decrease with particle size (80-350 nm). Measured
GF at 90% RH were compared with these calculated from size-resolved inorganic compositions
measured offline, and discrepancies between measured and calculated GF were attributed to the
effects of organics contained (Meier et al., 2009). In addition, hygroscopic growth at 55% and 70%
RH was also explored for 30-400 nm aerosol particles (Meier et al., 2009), and GF at 55% and 70%
RH, compared to 90% RH, displayed similar dependence on particle size.
Wu and co-workers (Wu et al., 2016; Wu et al., 2017; Wang et al., 2018c) carried out
exntensive aerosol hygrosocpic growth measurements (at 90% RH) at the PKU site during 2014-
2015. Bimodal aerosol hygroscopicity distribution was observed in May-June 2014 (Wu et al.,
2016), dominated by the hydrophilic mode, and the average $\kappa$ appeared to increase with particle
size, from 0.160 at 50 nm to 0.280 at 250 nm. In addition, number fractions of aerosol particles in



the hydrophilic mode first increased with particle size up to 150 nm, and then did not show
significant change with further increase in particle size (Wu et al., 2016); to be more specific,
average number fractions of aerosol particles in the hydrophilic mode were ~0.6 at 50 nm and
increased to ~0.8 above 150 nm. For each particle size, aerosol hygroscopicity was found to be
larger during new particle formation (NPF) periods, compared to non-NPF periods (Wu et al.,
2016), because more secondary species were found during NPF periods typically associated with
strong photochemical processes. Aerosol mass spectrometry (AMS) measurements suggested that
both aerosol hygroscopicity was dominated by inorganics, the contribution of which increased
with particle size and pollution levels (Wu et al., 2016). It was further found that the measured $\kappa$
could be well predicted using the AMS data, and the derived $\kappa$ of organics depended linearly on
their O:C ratios (Wu et al., 2016).
The PKU site was affected by a series of biomass burning events in May-June 2014, and the
effect of biomass burning on aerosol composition and hygroscopicity was examined (Wu et al.,
2017). During biomass burning events, biomass burning contributed significantly to the production
and growth of aerosols in the Aitken mode, and the contribution of organics and black carbon to
mass concentrations of submicrometer aerosols reached 60% and 18% (Wu et al., 2017).
Hygroscopicity and number fractions of aerosols in the hydrophobic mode were relatively
invariable during biomass burning events, and the average $\kappa$, which showed no variation with
particles size (50-250 nm), were determined to be ~0.1 (Wu et al., 2017), substantially smaller than
those in the same period without significant impacts by biomass burning (Wu et al., 2016).

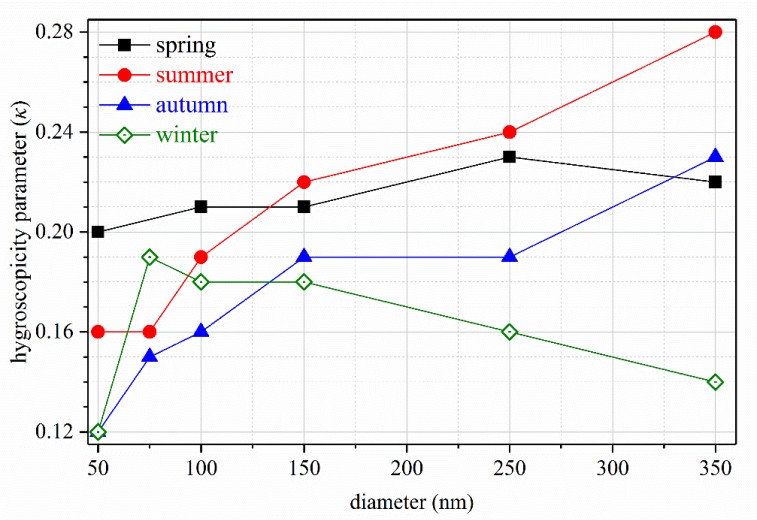

**Figure 1.** Change in average $\kappa$ with aerosol diameter at the PKU site in four different seasons between May 2014 to January 2015 (Wang et al., 2018c).

Seasonal variation of aerosol hygroscopic growth was investigated at the PKU site from May 2014 to January 2015 (Wang et al., 2018c), and the result is displayed in Figure 1. Average $\kappa$ increased significantly with particle size (50-350 nm) in summer and autumn, when strong photochemical processes enhanced secondary aerosol formation and led to particle growth (Wang et al., 2018c); in fact, number fractions of particles in the hydrophilic mode increased with pollution levels, and they dominated the accumulation mode when $PM_{2.5}$ mass concentration exceeded 100 $\mu g/m^3$. In contrast, as shown in Figure 1, average $\kappa$ only increased slightly with particles size (50-350 nm) in spring while decreased substantially with particle size (75-350 nm) in winter (Wang et al., 2018c), indicating significant contribution of primary species to aerosol particles. Furthermore, being different to summer and autumn, substantial amounts of aerosol particles in the hydrophobic mode were always observed in spring and winter (Wang et al., 2018c). Another important feature revealed by Figure 1 is that for 150-350 nm aerosols, the hygroscopicity





was always highest in summer and lowest in winter (Wang et al., 2018c), and the difference
between the two seasons increased with particle size.
In addition, aerosol hygroscopic growth was investigated in March-April 2015 at the roof of
the Environmental Science Building (40º0'17''N, 116º19'34''E) on the campus of Tsinghua
University (Fajardo et al., 2016). This site, very close to the PKU site, is usually affected by the
same air masses. Number size distributions under dry and ambient conditions were measured for
10-500 nm particles to explore aerosol hygroscopicity under ambient RH (Fajardo et al., 2016).
No obvious aerosol growth was observed for RH below 50% (Fajardo et al., 2016); however, the
aerosol volume was increased by ~80% when RH reached 50%, and further increase in ambient
RH led to further hygroscopic growth.
**IAP site:** The IAP site is located at the Institute of Atmospheric Physics, Chinese Academy
of Science (39.97ºN, 116.37ºE) between the third and fourth ring roads in northern Beijing. All the
aerosol hygroscopic growth measurements (Wang et al., 2017c; Wang et al., 2019b; Fan et al.,
2020; Jin et al., 2020) were conducted at 90% RH at the ground level.



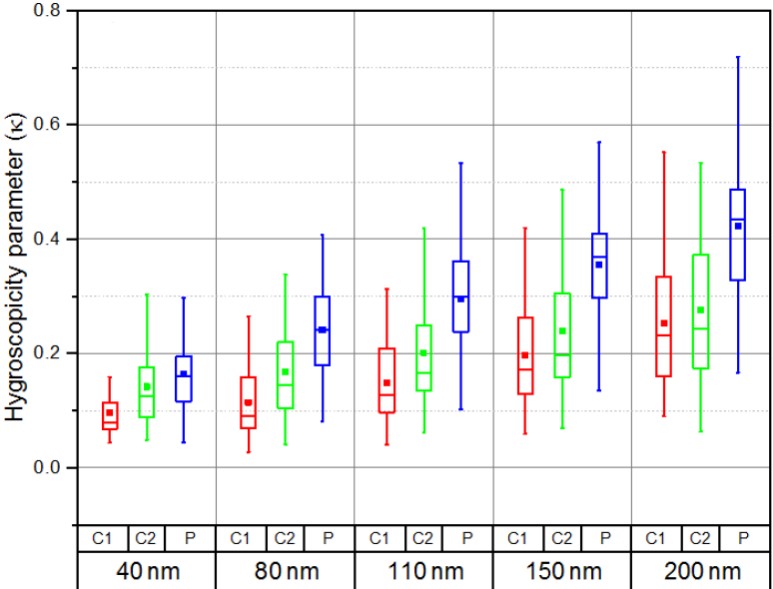


**Figure 2.** Size-resolved $\kappa$ during the control clean (C1), the non-control clean (C2) and the non-

control polluted (P) periods. Solid squares represent the average $\kappa$, boxes represent the 25th, 50th,

and 75th percentiles, and extremities represent the 5th and 95th percentiles. Reprint with

permission by Wang et al. (Wang et al., 2017c). Copyright 2017 Copernicus Publications.


Wang et al. (Wang et al., 2017c) investigated aerosol hygroscopic growth at the IAP site in
August-October 2015, when emission control measures were implemented for the 2015 China
Victory Day parade. Three periods with different pollution levels, including the control clean (C1),
the non-control clean (C2) and the non-control polluted (P) periods, were specifically examined to
evaluate the effect of emission control. Figure 2 shows that aerosol hygroscopicity increased with
particle size and pollution level (Wang et al., 2017c), due to enhanced contribution of secondary
species. For example, $\kappa$ increased from 0.100 at 40 nm to 0.250 at 200 nm during C1, from 0.140
at 40 nm to 0.280 at 200 nm during C2, and from 0.160 at 40 nm to 0.420 at 200 nm during the



polluted period (Wang et al., 2017c). Furthermore, number fractions of particles in the more
hygroscopic mode increased in the polluted period, compared to C1 and C2. For 40 nm particles,
a quasi-unimodal hygroscopicity distribution was observed during C1, while bimodal or quasi-
trimodal distributions were observed during the other two periods; in contrast, bimodal patterns
were always observed for 150 nm particles (Wang et al., 2017c). It was also found that for all the
three periods, the average $\kappa$ were always larger during the daytime than the nighttime (Wang et al.,
2017c).

A following study (Wang et al., 2019b) measured aerosol hygroscopic growth at the IAP site

in November-December 2016. Overall the average $\kappa$ were found to increase with particle size,
from 0.164 at 40 nm to 0.230 at 200 nm during the clean period and from 0.155 at 40 nm to 0.290
at 200 nm during the polluted period (Wang et al., 2019b); compared to the clean period, the
average $\kappa$ during the polluted period were smaller for 40 nm particles but larger for 80-200 nm
particles. In addition, bimodal distributions were always observed (Wang et al., 2019b). Number
fractions of particles in the less-hygroscopic mode was larger for 40 nm particles and smaller for
80-200 nm particles during the polluted period (Wang et al., 2019b), when compared to the clean
period, reflecting the compositional variation in 40 and 80-200 nm particles during the two periods.
Diurnal variation of aerosol hygroscopicity was also explored, displaying significant differences
between clean and polluted periods (Wang et al., 2019b).

Jin et al. (Jin et al., 2020) further analyzed size-resolved aerosol composition and

hygroscopicity measured at the IAP site in November-December 2016 (Wang et al., 2019b). The
size-dependent $\kappa$ derived from measured GF at 90% RH was used to calculate ALWC at ambient
RH, assuming that a constant $\kappa$ could be used to calculate GF at different RH (Jin et al., 2020); in
addition, size-resolved aerosol composition measured using AMS was used as input in



ISORROPIA-II to simulate ALWC at ambient RH. ALWC simulated using ISORROPIA-II were

found to be significantly smaller than calculated ALWC when RH was <60% (Jin et al., 2020),

because ISORROPIA-II failed to estimate water uptake by organics at low RH. Overall, organic

materials were estimated to contribute to (30±22)% of ALWC (Jin et al., 2020), highlighting the

importance of organics to aerosol hygroscopicity in urban Beijing.

Fan et al. (Fan et al., 2020) further conducted aerosol hygroscopic growth measurements at

the IAP site in May-June 2017, and bimodal hygroscopicity distributions were also observed for

40-200 nm aerosols. The summertime measurement in 2017 was compared with the wintertime

measurement at the same site in 2016 (Wang et al., 2019b), and the size dependence of aerosol

hygroscopicity was found to differ for the two seasons (Fan et al., 2020). The average $\kappa$ increased

from 0.158 at 40 nm to 0.271 at 110 nm in winter, and further increase in particle size (to 200 nm)

led to slight decrease in $\kappa$ (Fan et al., 2020); for comparison, the average $\kappa$ increased with particles

size in summer, from 0.211 at 40 nm to 0.267 at 200 nm (Wang et al., 2019b). It was suggested

that the size dependence of aerosol hygroscopicity was mainly determined by the size-resolved

mass fractions of secondary inorganic species (Fan et al., 2020).

**CAMS site:** Wang et al. (Wang et al., 2018a) measured aerosol hygroscopic growth (30-90%

RH) of ambient aerosols on the campus of Chinese Academy of Meteorological Sciences, located

between the second and third ring roads in west Beijing. Measurements were conducted on a

building roof (~53 m above ground level) in December 2016, and the distance between the site

and a major road with heavy traffic was <200 m. Aerosol hygroscopic growth displayed unimodal

when RH did not exceed 60%, while bimodal distributions were usually observed at 70% and 80%

RH; in addition, aerosol hygroscopic growth occasionally exhibited trimodal distribution at 85%

and 90% RH (Wang et al., 2018a). Measured GF at 90% RH were used to calculate $\kappa$, which were



determined to be 0.010-0.015 and 0.286-0.358 for the hydrophobic and hydrophilic modes (Wang
et al., 2018a), both increasing with particle size (50-200 nm). Number fractions of hydrophobic
particles exceeded 50% at 50 and 100 nm, while hydrophilic particles frequently became dominant
in terms of number concentrations at 150 and 200 nm (Wang et al., 2018a). In addition,
hygroscopicity decreased at 50 nm but increased at 200 nm during heavily polluted periods (Wang
et al., 2018a), indicating their difference in compositions and sources.
**3.1.2   Rural sites in Beijing**

Aerosol hygroscopic growth were measured at two rural sites in Beijing, including Yufa

(Achtert et al., 2009) and Huairou (Wang et al., 2020b). The Yufa site (39.51ºN, 116.31ºE) is ~1.2
km away from a high-traffic expressway and ~50 km south to urban Beijing, and can be considered
as a representative rural and regional background site. Achtert et al. (Achtert et al., 2009) measured
aerosol hygroscopic growth as a function of RH (56, 76, 85 and 91%) on a four-floor building (22
m above the ground) at this site in August-September 2006. GF at 91% RH, ranging from 1.15 to
1.80 for 30-300 nm particles, were found to be larger in the accumulation mode than the Aitken
mode (Achtert et al., 2009); furthermore, increase in mass fractions of sulfate during polluted
periods led to increase in aerosol hygroscopicity with pollution level. Diurnal variation of aerosol
hygroscopicity was also explored (Achtert et al., 2009): hygroscopicity was found to be higher in
the daytime than the nighttime for the Aitken mode, whereas no significant difference in
hygroscopicity was observed between daytime and nighttime for the accumulation mode.

The Huairou site (40.42ºN, 116.69ºE) is located on the campus of the University of the

Chinese Academy of Sciences, ~60 km northeast from the center of Beijing. It was mainly
influenced by regional transport of pollutants from downtown Beijing (Tan et al., 2018) and small
local sources nearly (e.g., moderate traffic and small residential areas). Aerosol hygroscopic



growth (at 90% RH) was measured at this site in January-March 2016 (Wang et al., 2020b). The
average $\kappa$ were determined to be 0.162-0.208 for 50-300 nm particles (Wang et al., 2020b), and
mass fractions of nitrate, which contributed significantly to aerosol hygroscopic growth, reached
44% during polluted episodes.
**3.1.3    Other urban/suburban sites**
Aerosol hygroscopic growth was measured at other four urban/suburban sites in NCP,
including two sites in Tianjin, one site in Hebei Province and one site in Shanxi Province.
**Tianjin:** The Wuqing site is located next to the Wuqing Meteorological Station (39º23'N,
117º0'E) in the west area of Wuqing (Tianjin), surrounded by mixed agricultural, residential and
industrial regions. This site is a good place to study regional air pollution in NCP, as it is ~30 km
northwest to the urban Tianjin, ~80 km southeast to the urban Beijing, ~130 km southwest to
Tangshan (Hebei), and ~160 km northeast to Baoding (Hebei). Aerosol hygroscopic growth was
measured at three RH (90%, 95% and 98.5%) at this site in July-August 2009 (Liu et al., 2011).
Bimodal hygroscopicity distribution was observed over the whole period, and the average $\kappa$,
derived from GF measured at 90% RH, increased from 0.250 at 50 nm to 0.340 at 250 nm (Liu et
al., 2011). Compared to the nighttime, both the average $\kappa$ and number fractions of particles in the
more-hygroscopic mode were larger during the daytime (Liu et al., 2011). The average $\kappa$ were
found to increase with particle size for the more-hygroscopic mode, from 0.310 at 50 nm to 0.390
at 250 nm (Liu et al., 2011); in contrast, they decreased with particle size for the nearly
hydrophobic mode, from 0.054 at 50 nm to 0.025 at 250 nm. It was found that inorganics play an
important role for hygroscopic growth of the accumulation mode (Liu et al., 2014), while organics
were very importance for hygroscopic properties of the Aitken mode. In addition, $\kappa$ calculated





from aerosol compositions measured offline were consistent with those derived from H-TDMA
measurements (Liu et al., 2014).

Two different methods were used to estimate ALWC at the Wuqing site in July-August 2009

(Bian et al., 2014). For the first method, $\kappa$ derived from GF measurements at 90-98.5% RH were
assumed to be constant at different RH, and thus ALWC could be calculated from particle number
size distribution (Bian et al., 2014); for the second method, size-resolved aerosol composition,
only taking into account water soluble inorganic ions, was used as input in ISORROPIA-II to
predict ALWC. ALWC estimated using the first method agreed with those using the second method
for >60% RH, but was much larger compared to the second method when ambient RH was <60%
(Bian et al., 2014).

In March 2018, Ding et al. (Ding et al., 2019) carried out aerosol hygroscopic growth

measurements (70-85% RH) at the NKU site, an air quality research supersite at Nankai University
(38º59'N, 117º20'E), which was ~20 km away from downtown Tianjin. GF measured at 85% RH
were used to calculate average $\kappa$, being 0.301-0.477, 0.203-0.386 and 0.281-0.419 on 13th, 14th
and 15th March (Ding et al., 2019). In addition, the average $\kappa$ were found to be larger during
polluted periods than clean periods, as the contribution of nitrate, sulfate and ammonium in the
accumulation mode increased during polluted periods (Ding et al., 2019). It was also found that
for the accumulation mode, $\kappa$ were larger in the nighttime than the daytime (Ding et al., 2019).
Water-soluble inorganic ions measured offline were used as input in the ISORROPIA-II to predict
aerosol hygroscopicity, and measured and predicted $\kappa$ showed good agreement (Ding et al., 2019),
implying that the contribution of organics to aerosol hygroscopic growth was quite limited.

**Hebei Province:** The Xingtai site is located at the National Meteorological Basic Station in

Xingtai (37.18ºN, 114.37ºE), a heavily polluted city in the center of NCP, and aerosol hygroscopic

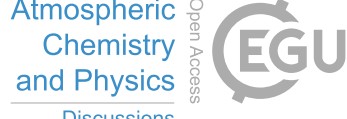

growth (at 85% RH) was measured at this site in May-June 2016 (Wang et al., 2018b). As shown
in Figure 3, quasi-unimodal aerosol hygroscopicity distribution was observed and number
fractions of particles in the more-hygroscopic mode was ~90% for 40-200 nm particles (Wang et
al., 2018b), indicating that they were highly aged and internally mixed. The average $\kappa$ were
determined to be 0.364-0.39 (Wang et al., 2018b), significantly larger than those reported for most
of other sites in NCP. No obvious dependence of average $\kappa$ on particle size was observed, and the
average $\kappa$ were found to be larger in daytime than nighttime, especially during new particle
formation events.

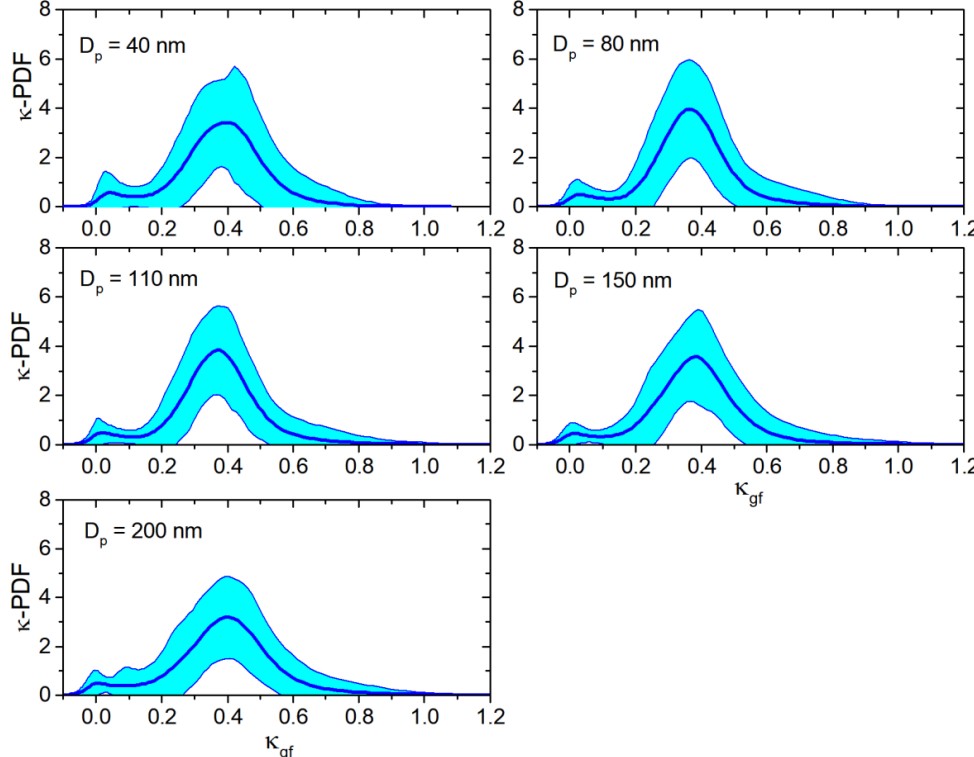


**Figure 3.** Mean probability density functions of $\kappa$ and their standard deviations (shaded areas) for

40, 80, 110, 150 and 200 nm particles at the Xingtai site in May-June 2016, as derived from





measured GF at 85% RH. Reprint with permission by Wang et al. (Wang et al., 2018b). Copyright
2018 Copernicus Publication.

For the campaign at the Xingtai site in May-June 2016 (Wang et al., 2018b), aerosol

hygroscopicity on a clean day (21 May) was compared with that on a highly polluted day (23 May).
Aerosol hygroscopicity was higher on the polluted day (Chen et al., 2019), likely due to the
enhanced formation of nitrate as revealed by ACSM (aerosol chemical speciation monitor)
measurements. Furthermore, aerosol hygroscopicity increased with particles size (40-200 nm) on
both days, from 0.288 to 0.339 on 21 May and from 0.325-0.352 on 23 May (Chen et al., 2019).

**Shanxi Province:** The Xinzhou site (38.24ºN, 112.43ºE, 1500 m above sea level) was located

on the border between the NCP and the Loess Plateau. This suburban and regional site, surrounded
by agricultural land with limited local anthropogenic emissions, was located ~360 km southwest
to Beijing, ~78 km northwest to Taiyuan and ~10 km south to the city nearby. Aerosol hygroscopic
growth (85% RH) was investigated for 25-200 nm aerosols at this site in July-August 2014 (Zhang
et al., 2017). Quasi-unimodal aerosol hygroscopicity distribution was observed, indicating highly
aged and internally mixed particles. The average $\kappa$ were determined to be 0.420-0.528,
significantly larger than those observed at other sites in the NCP; in addition, no obvious
dependence of $\kappa$ on particle size was found (Zhang et al., 2017).
**3.1.4    Other rural sites**

Aerosol hygroscopic growth was measured at other two rural sites in NCP, i.e. the Xianghe

site and the Wangdu site (both in Hebei). The Xianghe site (39.75ºN, 116.96ºE), surrounded by
residential areas and farmlands, is considered as a typical rural site in NCP and is located ~5 km
west to the center of Xianghe town and ~70 km southeast to Beijing. At this site, aerosol



hygroscopic growth (at 87% RH) was measured in July-August 2013 (Zhang et al., 2016b).
Trimodal aerosol hygroscopicity distributions were observed for 50-350 nm particles (Zhang et al.,
2016b), and the average $\kappa$ were determined to be 0.020-0.056, 0.170-0.211 and 0.365-0.455 for
nearly-hydrophobic, less-hygroscopic and more-hygroscopic modes. Aerosol hygroscopicity
showed some dependence on air masses (Zhang et al., 2016b): air masses which were transported
from the north with high speed winds typically contained larger number fractions of hydrophobic
species and exhibited lower hygroscopicity, whereas no obvious difference in aerosol
hygroscopicity and mixing state were observed for other air masses.
The Wangdu site (38.71ºN, 115.16ºE), a rural site located in the center area of NCP, was ~200
km southwest to Beijing, and aerosol hygroscopic growth (at 90% RH) was measured at this site
in June 2014 (Wang et al., 2017b). Bimodal aerosol hygroscopicity distribution was always
observed (Wang et al., 2017b), and the average $\kappa$ were found to increase with particle size, from
0.240 at 30 nm to 0.320 at 250 nm.
**3.2  Yangtze River Delta (YRD)**
A number of aerosol hygroscopic growth measurements have been carried out since 2009 in
three large cities (Shanghai, Hangzhou and Nanjing) in the Yangtze River Delta.
**3.2.1    Shanghai**
Ambient aerosol hygroscopic growth was measured at two sites in Shanghai (Ye et al., 2011;
Ye et al., 2013; Wang et al., 2014; Xie et al., 2017; Li et al., 2018; Wang et al., 2020a). The FDU
site (31°18'N, 121°29'E) is located on the building roof of Department of Environmental Science
and Engineering, Fudan University; the Pudong site (31.22ºN, 121.55ºE) is located in Pudong
Meteorological Bureau. Both sites are considered as urban sites, surrounded by residential,
industrial and traffic areas, and their distance is <10 km.





**FDU site:** At the FDU site, Ye et al. (Ye et al., 2011) measured aerosol hygroscopic growth

(30-200 nm) at 20-85% RH in January-February 2009. Bimodal hygroscopic growth distribution

was always observed at 85% RH, and $\kappa$ derived from measured GF at 85% RH were determined

to be 0.027-0.063 and 0.291-0.381 for the less- and more-hygroscopic modes (Ye et al., 2011). The

average $\kappa$ decreased with particle size for the less hygroscopic mode while increased with particle

size for the more hygroscopic mode (Ye et al., 2011); in addition, number fractions of particles in

the less hygroscopic mode decreased with particle size. The change in GF with RH (20-85%) was

also discussed for particles with different sizes (Ye et al., 2011).

Compositional data provided by ATOFMS (Aerosol Time-of-Flight Mass Spectrometry) were

used to interpret GF measured at 85% RH for 250 nm particles on 18-19 January and 10 February

2009 (Ye et al., 2011). Bimodal aerosol hygroscopicity distribution was observed for 250 nm

particles, including a nearly-hydrophobic mode with $\kappa$ of 0.029-0.061 and a more-hygroscopic

mode with $\kappa$ of 0.387-0.399 (Wang et al., 2014). Aerosols in the more-hygroscopic mode consisted

predominantly of secondary species (e.g., OC-amine, sulfate and nitrate), while biomass burning

aerosols, uncoated EC, secondary organic compounds, and dust/ash were frequently identified in

the nearly-hydrophobic mode (Wang et al., 2014).

Aerosol hygroscopic growth (at 85% RH) was also measured at this site in February-March

2014 (Wang et al., 2020a). Aerosol hygroscopicity was found to exhibit bimodal distribution at

250 nm, and the average $\kappa$ were determined to be 0.029 and 0.376 for nearly-hydrophobic and

more-hydrophilic modes (Wang et al., 2020a). Nearly-hydrophobic particles typically included

biomass burning aerosol, fresh EC and high molecular mass OC, while more-hydrophilic particles

included aged EC, amine-rich particles, and etc. (Wang et al., 2020a). Furthermore, a statistic



method was developed to estimate aerosol hygroscopicity from single particles mass spectra

(Wang et al., 2020a).

Xie et al. (Xie et al., 2017) further measured aerosol hygroscopic growth (83% RH) at the FDU

site in December 2014-January 2015. Bimodal aerosol hygroscopicity distribution (nearly

hydrophobic and more hygroscopic modes) was usually observed, and the average $\kappa$ increased

from 0.161 at 40 nm to 0.345 at 400 nm (Xie et al., 2017). Number fractions of nearly hydrophobic

particles increased during polluted periods for all the sizes considered (40-400 nm), indicating

significant contribution of primary particles during haze events (Xie et al., 2017); however, the

increase in number fractions of nearly hydrophobic particles during pollution events were less

significant for larger particles, suggesting that primary emissions contributed more to smaller

particles.

Mixing state and hygroscopic growth (at 85% RH) were explored at the FDU site in July 2017

specifically for ambient black carbon (BC) aerosols (120, 240 and 260 nm) (Li et al., 2018).

Number fractions of BC particles decreased with particle size, from ~80% for 120 nm to ~60% for

360 nm. Hygroscopicity of BC particles displayed unimodal distribution, and their GF at 85% RH

peaked at ~1.0 (Li et al., 2018). Enhancement in hygroscopicity of BC particles, due to their aging

via condensation of secondary species, was frequently observed (Li et al., 2018): during the

nighttime nitrate contributed significantly to BC aging, while formation of secondary organic

materials played an important role during the daytime.

**Pudong site:** Aerosol hygroscopic growth (at 91% RH) was studied at the Pudong site in

September 2009 (Ye et al., 2013). As shown in Figure 4, aerosol hygroscopicity was found to be

trimodal, including a nearly-hydrophobic mode and a more-hygroscopic mode, as well as a less-

hygroscopic mode with much less abundance (Ye et al., 2013). The average $\kappa$ increased from 0.270





at 30 nm to 0.390 at 200 nm for the more-hygroscopic mode (Ye et al., 2013), and decreased from
0.054 at 30 nm to 0.011 at 200 nm for the nearly-hydrophobic mode.

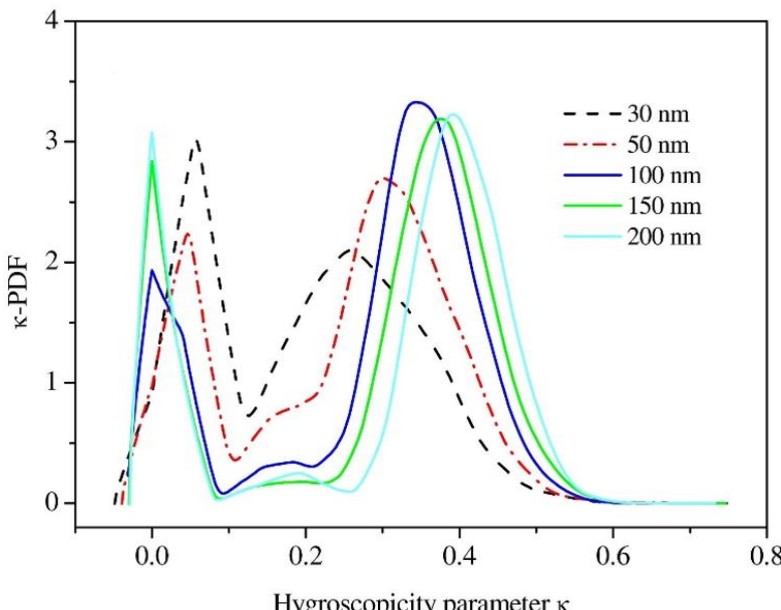


**Figure 4.** Probability distribution functions of the hygroscopicity parameter ($\kappa$) for 30, 50, 100,
150 and 200 nm aerosols at the Pudong site in September 2009. Reprint with permission by Ye et
al. (Ye et al., 2013). Copyright 2013 Elsevier Ltd.

**3.2.2   Hangzhou**

Up to now only one aerosol hygroscopic growth study was carried out in Hangzhou, at the ZJU

site located on the Huajiachi campus of Zhejiang University (30°16'N, 120°11'E). Aerosol
hygroscopic growth was measured at 70-90% RH (mainly at 82%) in December 2009-January
2010 (Zhang et al., 2011). Bimodal hygroscopicity distribution was observed for 50-200 nm
aerosols, while unimodal hygroscopicity distribution was observed for 30 nm aerosols (Zhang et
al., 2011). The average $\kappa$ decreased from 0.121 at 30 nm to 0.065 at 80 nm for the low-hygroscopic





mode, and further increase in particle size (up to 200 nm) did not lead to significant change in $\kappa$
(Zhang et al., 2011). For comparison, the average $\kappa$ increased from 0.303 at 30 nm to 0.343 at 80
nm for the more-hygroscopicity mode, and further increase in particle size only resulted in very
small increase in $\kappa$. In addition, number fractions of particles in the more-hygroscopic mode
increased from ~48% at 30 nm to ~70% at 100 nm, and remained nearly constant for 100-200 nm
(Zhang et al., 2011).
**3.2.3    Nanjing**

Aerosol hygroscopic growth was measured at three urban/suburban sites in Nanjing. The

NUIST site (32°207'N, 118°717'E) is a suburban site located on the 12[th] floor of the Meteorological
building at Nanjing University of Information Science and Technology, with several large
petrochemical factories and a busy expressway nearby. The NATC site (32.0°N, 118.7°E) is a
typical urban site at Nanjing Advanced Technical College, located in the centre business district
with heavy residential and traffic emissions. The JEMC site is an urban site on the 6[th] floor of the
building of Jiangsu Environmental Monitoring Centre (~18 m above the ground), located in the
urban area and surrounded by a variety of sources such as residence, restaurants, office blocks and
traffic.

**NUIST site:** Wu et al. (Wu et al., 2014) measured aerosol hygroscopic growth as a function of

RH (60-90%) at the NUIST site in May-July 2012, and bimodal hygroscopicity distributions were
frequently observed at 90% RH for 40-200 nm aerosols. For the more-hygroscopic mode, $\kappa$ were
determined to be 0.294-0.349, increasing with particle size (except for 40 nm); while for the less-
hygroscopic mode, $\kappa$ were found to decrease with particle size, from 0.079 at 40 nm to 0.040 at
200 nm (Wu et al., 2014). The average aerosol hygroscopicity measured at this site in Nanjing
seemed to be slightly lower than those reported in Beijing, Shanghai and Guangzhou.





Yang and co-workers further investigated aerosol (30-230 nm) hygroscopic growth (at 90%
RH) at this site in April-May 2014 (Xu et al., 2015; Yang et al., 2019), and bimodal hygroscopicity
distribution was observed. The $\kappa$ values were found to be very low (close to 0) for the low-
hygroscopic mode, and decreased from 0.232 at 30 nm to 0.186 at 230 nm for the medium
hygroscopic mode. Aerosol hygroscopicity measured in April-May 2014 (Xu et al., 2015; Yang et
al., 2019) were significantly lower than that measured in May-June 2012 at the same site (Wu et
al., 2014). One possible reason was that in April-May 2014 organic species made a large
contribution to submicrometer aerosols (21-38% by mass) (Xu et al., 2015; Yang et al., 2019), thus
leading to substantial decrease in aerosol hygroscopicity.
**NATC site:** In August 2013, Li et al. (Li et al., 2015b) investigated hygroscopic growth at 90%
RH for 32-350 nm aerosols. A less-hydrophobic mode ($\kappa$: 0.017-0.031) and a more-hygroscopic
mode ($\kappa$: 0.178-0.229) were observed during the campaign (Li et al., 2015b). Aerosol
hygroscopicity reported at the NATC site in August 2013 (Li et al., 2015b) was lower than these
reported at the NUIST site in May-June 2012 (Wu et al., 2014) and in April-May 2014 (Xu et al.,
2015; Yang et al., 2019), perhaps because the contribution of low hygroscopic primary particles
(e.g., soot) from local emission was larger at the NATC site (an urban site), compared to the
NUIST site (a suburban site).
**JEMC site:** At the JEMS site, 40-200 nm aerosol hygroscopic growth was measured at 85%
RH in January-February 2015 (Zhang et al., 2018). The average $\kappa$ were determined to be 0.200-
0.271 for 40-200 nm particles (Zhang et al., 2018), significantly larger than those (0.081-0.126 for
32-350 nm particles) reported for the NATC site in August 2013 (Li et al., 2015b), and the reason
was unclear. Bimodal hygroscopicity distribution was also observed (Zhang et al., 2018); similar
to two previous studies in Nanjing (Wu et al., 2014; Li et al., 2015b), number fractions of particles



in the low hygroscopic mode and their average $\kappa$ both decreased with particle size, while the
average $\kappa$ increased with particle size for the more hygroscopic mode (except for 40 nm).
**3.3 Pearl River Delta (PRD)**
A series of aerosol hygroscopic growth studies were conducted in PRD, to be more specific, at
two rural sites (Xinken and Wanqinsha) and one suburban site (Panyu) in Guangzhou and one
suburban site (HKUST) in Hong Kong.
**3.3.1    Rural sites in Guangzhou**
The Xinken site (22.6°N, 113.6°E), located near the Pearl River estuary, is ~50 km southeast
to urban Guangzhou, and the Wanqinsha site is located ~9 km northwest of Xinken. Both are
typical rural background sites with no major pollution sources nearby, and air quality at both sites
are affected by regional transport combined with limited local sources, such as traffic, ships,
biomass burning and cooking (Cheng et al., 2006; Eichler et al., 2008; Kim et al., 2011).
Eichler et al. (Eichler et al., 2008) measured aerosol hygroscopic growth (30-91% RH) at the
Xinken site in October-November 2004. The average GF at 91% RH were determined to 1.45,
1.53, 1.6 and 1.56 for 80, 140, 250 and 380 nm particles (Eichler et al., 2008), corresponding to $\kappa$
of 0.244, 0.283, 0.324 and 0.288, respectively. Inorganic aerosol compositions measured offline
were used to calculate GF, and the average difference between the measured and calculated GF
was found to be <8% (Eichler et al., 2008), suggesting that the contribution of organics to aerosol
hygroscopicity was rather small.
In a following study (Kim et al., 2011), aerosol hygroscopic growth (at 85% RH) of ultrafine
particles (40, 50, 60 and 80 nm) was investigated at the Wanqinsha site in October-November 2008.
During photochemical events, GF varied between 1.13 and 1.55, and particles consisted mainly of
ammonium sulfate and organic materials (Kim et al., 2011). For comparison, during combustion





events (i.e. affected by biomass burning and traffic emission), aerosol particles were mainly
composed of non-hygroscopic carbonaceous species and smaller amounts of potassium, and
correspondingly measured GF were reduced to 1.05-1.15 (Kim et al., 2011).
**3.3.2   Urban/suburban sites in Guangzhou**
The Panyu site, located at the top of Mt. Dazhengang (23º00'N, 113º21'E, 150 m above the sea
level), is surrounded by residential areas without major pollution sources nearby and can be
considered as a suburban site in Guangzhou (Tan et al., 2013). Several aerosol hygroscopic growth
measurements at 90% RH have been carried out at this site since 2011 (Tan et al., 2013; Jiang et
al., 2016; Cai et al., 2017; Tan et al., 2017; Cai et al., 2018; Hong et al., 2018; Liu et al., 2018).
Aerosol hygroscopic growth was first measured at this site in November-December 2011 (Tan
et al., 2013). Bimodal hygroscopicity distributions were observed for 40, 80, 110, 150 and 200 nm
particles, and $\kappa$ were determined to be 0.045-0.091 and 0.290-0.323 for the less- and more-
hygroscopic modes (Tan et al., 2013). In general, both hygroscopicity and number fractions
increased with particle size for the more-hygroscopic mode, whereas they both decreased with
particle size for the less-hygroscopic mode. Average hygroscopicity was found to be larger during
the daytime than the nighttime for both modes (Tan et al., 2013), and hygroscopicity and number
fractions of particles in the more-hygroscopic mode increased during polluted periods, when
compared to clean periods.
Jiang et al. (Jiang et al., 2016) compared aerosol hygroscopicity measured at this site between
winter (December 2012-January 2013) and summer (July-September 2013), and no obvious
difference in average $\kappa$ was found between the two seasons. Trimodal hygroscopicity distributions
were observed for 40-200 nm particles, and $\kappa$ were determined to be 0.290-0.339, ~0.15 and ~0.015
for more-, less- and non-hygroscopic modes (Jiang et al., 2016). Similar to the work by Tan et al.





(Tan et al., 2013), hygroscopicity and number fractions increased with particle size for the more-
hygroscopic mode, with no distinct difference between winter and summer (Jiang et al., 2016); for
the non-hygroscopic mode, hygroscopicity and number fractions both decreased with particle size,
and their number fractions were slightly lower in winter than in summer. Furthermore, the average
$\kappa$ were larger during daytime than nighttime for both seasons, and the diurnal variation was more
profound in summer (Jiang et al., 2016).

Tan et al. (Tan et al., 2017) measured aerosol hygroscopic growth in January-March 2014, and

$\kappa$ increased from 0.204 at 40 nm to 0.312 at 200 nm. The $\kappa$ values derived from GF measured at
90% RH were used to calculate ALWC under ambient conditions, and meanwhile aerosol inorganic
species measured were used as input in ISORROPIA-II to predict ALWC. Good agreement
between calculated and predicted ALWC were found for RH >70%, but significant differences
were found at <70% RH (Tan et al., 2017). Liu et al. (Liu et al., 2018) further explored aerosol
hygroscopic growth measured in February-March 2014 at this site, and found that he average $\kappa$
values increased from 0.261 at 80 nm to 0.323 at 200 nm. In addition, bimodal hygroscopicity
distribution was observed, and $\kappa$ increased from 0.382 at 80 nm to 0.432 at 200 nm for the more
hygroscopic mode (Liu et al., 2018).

Aerosol hygroscopic growth (at 90% RH) were further measured at this site in November-

December 2014 (Cai et al., 2017; Cai et al., 2018). Bimodal hygroscopicity distributions were
observed for 40-200 nm particles, and the average $\kappa$ increased with particle size, from 0.213 at 40
nm to 0.312 at 200 nm. The $\kappa$ values derived from size-resolved chemical compositions measured
using AMS were significantly lower than those derived from GF measurements (Cai et al., 2017;
Cai et al., 2018), probably because using a constant $\kappa$ value (0.1) may underestimate
hygroscopicity of aerosol organics.





Aerosol composition and hygroscopic growth at 90% RH were investigated at this site in
September-October 2016 (Hong et al., 2018), using an ACSM and a H-TDMA. Bimodal
hygroscopicity distributions were observed; the more-hygroscopic mode was dominant at 100 and
145 nm, while less- and more-hygroscopic modes were of similar magnitude at 30 and 60 nm
(Hong et al., 2018). The average aerosol hygroscopicity increased with particle size, and no
obvious diurnal variation was observed (Hong et al., 2018); however, aerosol hygroscopicity was
higher during the daytime for the less-hygroscopic mode while slightly lower in the afternoon for
the more-hygroscopic mode. Hygroscopicity closure analysis suggested that taking into account
the dependence of GF on composition for organics led to better agreement between measured and
calculated GF (Hong et al., 2018). It was further found that GF increased linearly with O:C ratios
for organics, and the derived GF appeared to be less sensitive to the changes of O:C ratios during
polluted periods.
**3.3.3    Hong Kong**
Since 2011, H-TDMA and online mass spectrometry were employed by Chan and co-workers
(Lopez-Yglesias et al., 2014; Yeung et al., 2014; Cheung et al., 2015) to investigate aerosol
composition and hygroscopic growth at the HKUST supersite (22º20'N, 114º16'E) on the east coast
of Hong Kong. It is a typical suburban and coastal site with no major pollution sources nearby.
Aerosol hygroscopic growth at 90% RH was first investigated at this site in 2011 (Yeung et al.,
2014), and bimodal aerosol hygroscopicity distributions were observed with a dominant more-
hygroscopic mode and a weak less-hygroscopic mode at 75, 100, 150 and 200 nm. The average $\kappa$
were determined to be 0.330-0.360 during May, 0.370-0.390 during the first half of September,
0.210-0.250 during the second half of September and 0.290-0.320 during November (Yeung et al.,




2014), caused by compositional variations in different air masses; however, no obvious
dependence of average $\kappa$ on particle size was found.
Number fractions of particles in the more-hygroscopic mode were always >0.8 (Yeung et al.,
2014), except for 75 nm particles in the second half of September (~0.45) which was dominantly
affected by continental air masses. When compared to maritime aerosols, hygroscopicity of
aerosols in the more-hygroscopic mode was substantially lower for continental aerosols which
contained larger proportions of organic materials (Yeung et al., 2014). Hygroscopicity closure
analysis suggested that using a constant GF (1.18) at 90% RH for organic materials, instead of
considering the dependence of GF on their oxidation degree, would lead to better agreement
between measured and calculated GF (Yeung et al., 2014), likely because inorganic species (such
as sulfate) contributed dominantly to the overall aerosol hygroscopicity during the entire campaign.
In addition, hygroscopic growth at the HKUST site was investigated as a function of RH (10-
90%) in 2011-2012 (Lopez-Yglesias et al., 2014; Cheung et al., 2015), and both hysteresis behavior
and continuous hygroscopic growth of ambient aerosols were observed.
**3.4 Other locations**
In addition to NCP, YRD and PRD, measurements of aerosol hygroscopic growth were also
conducted in other regions in China, as discussed below.
**Taipei:** Hygroscopic growth (15-90% RH) was investigated for 53, 82, 95 and 202 nm aerosols
at an urban site in Taipei (Taiwan Province) in October-December 2001 (Chen et al., 2003).
Bimodal hygroscopicity distribution was observed for all the particles at 90% RH: while $\kappa$ (0.049-
0.068) showed no obvious dependence on particle size for the less hygroscopic mode, they
increased from 0.274 at 53 nm to 0.422 at 202 nm for the more hygroscopic mode (Chen et al.,
2003). No obvious hygroscopic growth was observed at <45% RH (Chen et al., 2003), and bimodal





hygroscopic growth behavior appeared at ~76% RH for all the sizes (53-202 nm), becoming more
noticeable with further increase in RH.
**Mt. Huang:** Mt. Huang (30º08'N, 118º09'E) is located in the mountainous area of east China
with large forest coverages and limited anthropogenic activities. Aerosol hygroscopic growth at
50-85% RH was examined in September-October 2012 at the mountain foot (~464 m above the
sea level) and the mountain top (~1860 m above the sea level) (Wu et al., 2018a). No significant
particle growth was observed below 60% RH at both sites, and bimodal growth behavior appeared
at ~75% RH except 40 nm particles and became more evident at higher RH (Wu et al., 2018a).
Hygroscopicity was higher in the daytime than the nighttime for both modes. In addition,
hygroscopicity was slightly higher at the mountain foot than the mountain top for both modes
(except 200 nm particles in the more-hygroscopic mode) (Wu et al., 2018a); the reason was that
more secondary inorganic species were formed at the mountain foot due to human activities, while
on the mountain top the contribution of organics increased. Compared to NCP, YRD and PRD sites,
the overall aerosol hygroscopicity was lower at Mt. Huang (Wu et al., 2018a), as it is located in a
clean region with smaller fractions of secondary inorganic aerosols.
In July 2014 aerosol hygroscopic growth (at 85% RH) was further studied at the top of Mt.
Huang (Xu, 2015; Chen et al., 2016; Wang et al., 2016). The average $\kappa$ were determined to be
0.275, 0.266 and 0.290 at 70, 150 and 230 nm (Chen et al., 2016; Wang et al., 2016), in good
agreement with the previous study conducted at the same site in 2012 (Wu et al., 2018a). At a
given particle size, aerosol hygroscopicity was found to be higher in the daytime than the nighttime
(Chen et al., 2016; Wang et al., 2016); furthermore, aerosol hygroscopicity was higher for air
masses from northwest than those from southeast. The derived $\kappa$ depended positively on mass
fractions of inorganics and negatively on organics (Chen et al., 2016; Wang et al., 2016). In



addition, unimodal aerosol hygroscopicity distribution occurred with high frequency (47.5%)
during the campaign, and it also appeared more frequently in the afternoon with GF (at 85% RH)
in the range of 1.25-1.45 (Chen et al., 2016; Wang et al., 2016).
**Shouxian:** In June-July 2016, Qian et al. (Qian et al., 2017) studied hygroscopic growth (at
90% RH) of 50-250 nm aerosols at Shouxian National Climate Observatory (32º26'N, 116º48'E)
in east China, a rural site surrounded by farmlands at Shouxian, Anhui Province. Bimodal aerosol
hygroscopicity distribution was observed, and the average $\kappa$ increased with particle size, from
0.129 at 50 nm to 0.279 at 250 nm (Qian et al., 2017).
**East China Sea:** Total suspended particles were collected during a cruise over the East China
Sea (22-35ºN and 119-126ºE) in May-June 2014 and dissolved in deionized water. The resulting
solutions were atomized to generated aerosols, and their hygroscopic growth was then measured
at 5-90% RH (Yan et al., 2017). The average $\kappa$ was determined to be 0.88 for the whole cruise, and
the daytime average (0.81) was smaller than the nighttime average (0.95) (Yan et al., 2017), due
to less chloride loss in the nighttime. It is to be assessed to which extent aerosols generated by Yan
et al. (Yan et al., 2017) can actually mimic ambient aerosols.
**3.5 Summary**
Geographically speaking, almost all the aerosol hygroscopic growth studies were conducted in
east China, especially in NCP, YRD and PRD. Aerosol hygroscopic growth in other regions in
China remains to be explored, and measurements at rural and remote areas with limited
anthropogenic impacts are very scarce. In addition, previous measurements were mainly
performed at or close to the ground level, except these carried out on the top of Mt. Huang (Chen
et al., 2016; Wang et al., 2016; Wu et al., 2018a).



It can be concluded that submicrometer aerosols in China usually exhibit bimodal
hygroscopicity distribution (i.e. nearly-hydrophobic and more-hygroscopic modes). However,
trimodal distributions, with a medium-hygroscopic mode with limited importance, were also
reported by several studies (Massling et al., 2009; Meier et al., 2009; Ye et al., 2013; Jiang et al.,
2016; Zhang et al., 2016b; Wang et al., 2017c; Wang et al., 2018a), and quasi-unimodal
hygroscopicity distributions existed but were quite sparse (Chen et al., 2016; Wang et al., 2016;
Wang et al., 2017c; Zhang et al., 2017; Wang et al., 2018b).
For the more-hygroscopic mode, $\kappa$ usually increased with particle size, except for the
measurements carried out at HKUST site (Yeung et al., 2014) where no obvious dependence on
particle diameter was found. For the nearly-hydrophobic mode, $\kappa$ usually decreased with particle
size (Liu et al., 2011; Ye et al., 2011; Zhang et al., 2011; Tan et al., 2013; Ye et al., 2013; Wu et al.,
2014; Jiang et al., 2016; Zhang et al., 2016b; Qian et al., 2017; Zhang et al., 2018), though different
results were also reported in several studies (Chen et al., 2003; Massling et al., 2009; Meier et al.,
2009; Li et al., 2015b; Wang et al., 2018a; Wu et al., 2018a).
Average aerosol hygroscopicity, especially for the more-hygroscopic mode, usually increased
with pollution levels (Massling et al., 2009; Wu et al., 2016; Wang et al., 2017c; Wang et al., 2018a;
Chen et al., 2019; Ding et al., 2019; Wang et al., 2019b), attributed to increased mass fractions of
secondary inorganic aerosols. However, different results were also reported (Meier et al., 2009),
especially for particles at or below 50 nm (Achtert et al., 2009; Wang et al., 2018a; Wang et al.,
2019b) for which primary emissions could play an important role.
A few studies examined aerosol hygroscopic growth at different seasons (Massling et al., 2009;
Jiang et al., 2016; Wang et al., 2018c; Fan et al., 2020). No obvious difference in the overall aerosol
hygroscopicity was observed between summer and winter at the PKU site (Beijing) (Massling et





al., 2009), the IAP site (Beijing) (Fan et al., 2020) and the Panyu site (Guangzhou) (Jiang et al.,
2016). However, one study (Wang et al., 2018c) suggested that the overall hygroscopicity, and
especially hygroscopicity of 150-350 particles, was highest in summer and lowest in winter at the
PKU site (Beijing).

Diurnal variations of aerosol hygroscopic growth were also investigated. Most of these studies

suggested that aerosol hygroscopicity was generally higher in the daytime, compared to the
nighttime. For example, hygroscopicity was higher in the daytime than the nighttime for the Aitken
mode at the Yufa site (Beijing) in August-September 2006 (Achtert et al., 2009), while no
significant difference was found between daytime and nighttime for the accumulation mode. In
addition, aerosol hygroscopicity was found to be higher at the daytime than the nighttime at the
IAP site (Beijing) in August-October 2015 (Wang et al., 2017c), at the Wuqing site (Tianjin) in
July-August 2009 for the more hygroscopic mode (Liu et al., 2011), at the Xingtai site (Hebei) in
May-June 2016 (Wang et al., 2018b), at the Panyu site (Guangzhou) in November-December 2011
(Tan et al., 2013), December 2012-Janurary 2013 (Jiang et al., 2016) and July-September 2013
(Jiang et al., 2016), and at Mt. Huang in September-October 2012 (Wu et al., 2018a) and July 2014
(Chen et al., 2016; Wang et al., 2016). The underlying reason might be that photochemical
processes led to increased relative contribution of secondary aerosols. However, there are also
expectations. For example, $\kappa$ was larger in the nighttime than the daytime for the accumulation
mode at the NKU site (Tianjin) in March 2017 (Ding et al., 2019). In addition, no obvious diurnal
variation in average aerosol hygroscopicity was observed at the Panyu site (Guangzhou) in
September-October 2016 (Hong et al., 2018), though aerosol hygroscopicity was higher during the
daytime for the less hygroscopic mode and slightly lower in the afternoon for the more-
hygroscopic mode.





While aerosol hygroscopic growth measurements were typically carried out at a single RH at
around 90%, several studies also investigated aerosol hygroscopic growth as different RH (Chen
et al., 2003; Eichler et al., 2008; Achtert et al., 2009; Meier et al., 2009; Liu et al., 2011; Ye et al.,
2011; Zhang et al., 2011; Cheung et al., 2015; Wang et al., 2018a; Wu et al., 2018a). As shown in
Figure 5, for the measurement carried out at ZJU site (Hangzhou) in December 2009-January 2010
(Zhang et al., 2011), $\kappa$ derived from measured GF at different RH (73-88%) varied from 0.186 to
0.244 for 150 nm particles and from 0.184 to 0.252 for 200 nm particles. For the measurement
carried out at the Yufa site (Beijing) in August-September 2006 (Achtert et al., 2009), $\kappa$ were found
to decrease with increasing RH (56-91%) for 250 nm particles, varying from ~0.3 to ~0.8.
Considerable variations of $\kappa$ with RH were also reported in other studies (Chen et al., 2003; Meier
et al., 2009; Ye et al., 2011; Cheung et al., 2015). Therefore, it can be concluded that using a
constant $\kappa$ to describe aerosol hygroscopic growth at different RH may not always be proper. In
addition, during most H-TDMA measurements aerosols were first dried at low RH (typically <15%)
and then humidified to a given RH, and as a result these measurements could not simulate the
formation of supersaturated droplets which may exist even when RH was below the corresponding
deliquescence RH but above the efflorescence RH.




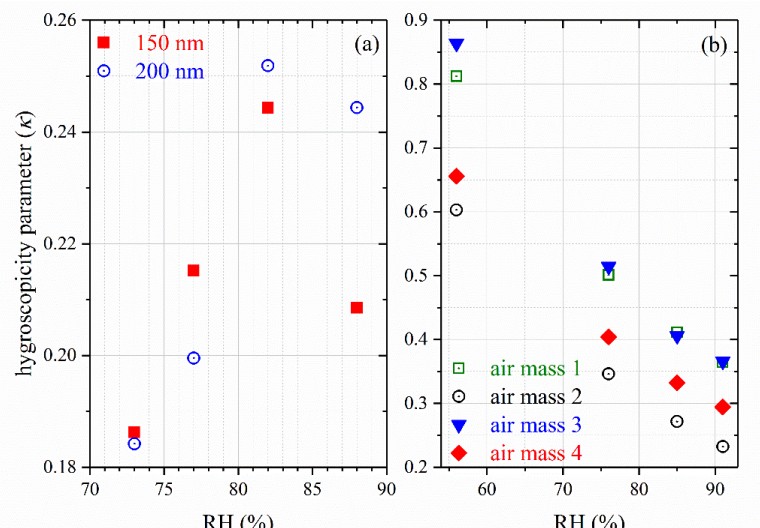

**Figure 5.** Single hygroscopicity parameters ($\kappa$) derived from GF measured as different RH. (a)
150 and 200 nm particles at the ZJU site (Hangzhou) in December 2009-January 2010 (Zhang et
al., 2011); (b) 250 nm particles at the Yufa site (Beijing) in August-September 2006 for four typical
air masses (Achtert et al., 2009).

## 4  CCN activities
As stated in Section 2.2, we only discuss CCN activity measurements which reported $\kappa$ values
herein.  Sections 4.1-4.3 review measurements conducted in NCP, YRD and PRD, and
measurements carried out in other regions in China are discussed in Section 4.4.
### 4.1  North China plain (NCP)
### 4.1.1    Beijing
In August-September 2006, size-resolved CCN activities were measured at the Yufa site
(Gunthe et al., 2011). Maximum activation fractions were around 1 for supersaturation in the range
of 0.26-0.86%; however, they only reached ~0.8 on average at 0.07% supersaturation, and these
inactive particles were mainly soot. For the entire measurement period, the average $\kappa_a$ and $\kappa_t$ were
both determined to be 0.3±0.1. CCN activities were found to increase with particle size due to
increased mass fractions of soluble inorganics (Gunthe et al., 2011), and $\kappa_a$ was measured to be
~0.2 at ~40 nm and ~0.5 at 200 nm. During periods affected by aged regional pollution, mass
fractions of soluble inorganics were enhanced, leading to increase in $\kappa_a$ (0.35±0.05) (Gunthe et al.,
2011); in contrast, mass fractions of organics increased during periods influenced by fresh city
pollution, resulting in decrease in $\kappa_a$ (0.22±0.07).

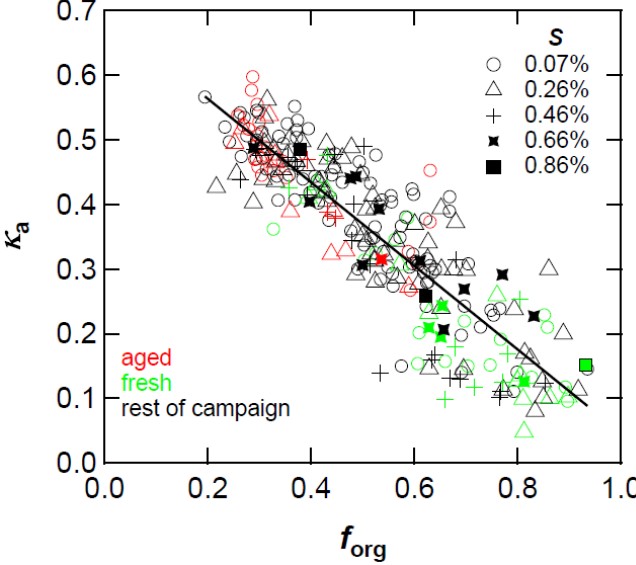


**Figure 6.** Dependence of $\kappa_a$ on mass fractions of organics for three periods over the campaign (red:
the aged regional pollution period; green: the fresh city pollution period; black: the rest of the
campaign). Reprint with permission by Gunthe et al. (Gunthe et al., 2011). Copyright 2011
Copernicus Publications.

As shown in Figure 6, the measured CCN activities decreased as mass fractions of organics

increased (Gunthe et al., 2011); furthermore, the measured $\kappa_a$ could be quantitatively described by





mass fractions of soluble inorganics and organics, and their $\kappa$ were determined to be 0.7 and 0.1.
Aerosol CCN activities during a rapid particle growth event on 23 August were further examined
(Wiedensohler et al., 2009), during which CCN size distribution was dominated by the growing
nucleation mode instead of the accumulation mode in usual.

Measurements were carried out at the PKU site to investigate size-resolved CCN activities in

May-June 2014 (Wu et al., 2017). Similar to the concurrent H-TDMA measurements, average $\kappa_a$
was determined to be ~0.10 during biomass burning events, displaying no dependence on particles
size (Wu et al., 2017). CCN activities of submicrometer particles were significantly reduced during
biomass burning periods, due to increased mass fractions of organics and black carbon.
Furthermore, average $\kappa$ calculated from aerosol compositions measured using AMS were
consistent with those derived from hygroscopic growth and CCN activity measurements (Wu et
al., 2017), if $\kappa$ were assumed to be 0.53 and 0 for inorganics and organics.

Zhang et al. (Zhang et al., 2017) investigated size-resolved CCN activities at the IAP site in

November-December 2014 and August-September 2015, and maximum activation fractions were
found to be much smaller than one, indicating large fractions of CCN-inactive particles from local
primary emissions. The average $\kappa_a$, which ranged from 0.22 to 0.31 for 60-150 nm particles and
increased with particle size (Zhang et al., 2017), agreed well with those derived from the
concurrent H-TDMA measurements (Wang et al., 2017c). In addition, $\kappa$ (0.32±0.11) calculated
using ACSM-measured aerosol composition were significantly larger than those derived from
hygroscopic growth (0.25±0.08) and CCN activities (0.26±0.04) (Zhang et al., 2017). This was
because hygroscopicity estimated using ASCM-measured composition did not consider the
contribution of smaller and less-hygroscopic particles (aerosol hygroscopicity became lower for
smaller particles, but ACSM only detected >60 nm particles).



In November-December 2016, Zhang et al. (Zhang et al., 2019a) further investigated size-
resolved CCN activities at the IAP site and found that [CCN] was significantly increased during
nucleation-initiated haze episodes. It was suggested that increase in particle size contributed >80%
to the observed increase in [CCN] (Ren et al., 2018; Zhang et al., 2019a), while the effect of aerosol
hygroscopicity enhancement, due to change in aerosol composition, was much smaller.
**4.1.2     Other locations in NCP**
Zhang et al. (Zhang et al., 2014; Zhang et al., 2017) measured size-resolved CCN activities at
the Xianghe site (39.75ºN, 116.96ºE) in June-July 2013. Average $\kappa_a$ were determined to be 0.24-
0.32 during polluted periods, showing no dependence on particle size; in contrast, $\kappa_a$ increased
from ~0.22 at ~50 nm to ~0.38 at ~180 nm for background days (Zhang et al., 2014). Compared
to polluted periods, $\kappa_a$ were ~20% larger under background conditions for the accumulation mode
(100-200 nm), as the contribution of aerosol organics from fresh biomass burning was significantly
increased during pollution events (Zhang et al., 2014); however, $\kappa_a$ were very similar for the
nucleation/Aitken modes (40-100 nm) under background and polluted conditions.
Size-resolved CCN activities were further investigated at Xianghe site in July-August 2013
(Ma et al., 2016; Tao et al., 2020), and it was found that $\kappa_a$ increased with particle size, from
0.22±0.02 at 46 nm to 0.38±0.02 at 179 nm. Compared to that derived from concurrent H-TDMA
measurements, aerosol hygroscopicity derived from CCN activities were slightly lower for <50
nm particles but higher for >100 nm particles (Ma et al., 2016; Zhang et al., 2016b).
Zhang and co-workers (Zhang et al., 2016a; Li et al., 2017b; Zhang et al., 2017) also
investigated size-resolved CCN activities at the Xinzhou site in July-August 2014. The average $\kappa_a$
were determined to be 0.42-0.51 for 37-150 nm particles, exhibiting no dependence on particle
size (Zhang et al., 2017); compared to other sites in the NCP, aerosols at the Xinzhou site displayed



significantly higher CCN activities, as aerosols observed at this site were highly aged after
undergoing regional transport for a long time. The average $\kappa_a$ (0.48±0.07) (Zhang et al., 2017)
agreed well with that (0.47±0.03) determined from concurrent H-TDMA measurements (Zhang et
al., 2017), both much significantly larger than that (0.41±0.06) calculated from ACSM-measured
aerosol composition, probably because such calculation may underestimate the hygroscopicity of
aerosol organics.
**4.2 Yangtze River Delta (YRD)**
Size-resolved CCN activity measurements were conducted in August 2013 at the NBM site
(32.04ºN. 118.70ºE) on the Jiangxi Island in the Yangtze River (Ma et al., 2017). This site, located
in a suburban area of Nanjing, did not have significant local emission at that time. The $\kappa_a$ values
were found to range from ~0.1 to ~0.8 during the campaign, being 0.35±0.13 on average (Ma et
al., 2017), and no significant variation in average $\kappa_a$ was found for biomass burning, urban, marine
and industrial air masses. In addition, $\kappa_a$ increased from 0.30±0.08 at ~55 nm to 0.34±0.08 at 67
nm, due to larger contribution of low-hygroscopic organics at 50 nm; however, further increase in
particle size up to ~149 nm did not lead to obvious increase in $\kappa_a$ (Ma et al., 2017), likely because
aerosols arriving at this site were heavily aged and well internally mixed.
Ling-term size-resolved CCN activities were studied in January-December 2013 at the Lin'an
site (Hangzhou, Zhejiang Province) (Che et al., 2016; Che et al., 2017), which is a WMO Global
Atmospheric Watch regional station (30.3ºN, 119.73ºE, 138 m above the sea level) located in the
center of YRD. Maximum activation fractions were close to one at high supersaturation but only
reached ~0.89 at 0.1% supersaturation. Values of $\kappa_a$ and $\kappa_t$ were almost identical (~0.25) at 40-50
nm and increased to ~0.42 ($\kappa_a$) and ~0.40 ($\kappa_t$) at 100-150 nm (Che et al., 2017), suggesting that
larger particles contained larger fractions of hygroscopic species (e.g., soluble inorganics).





Furthermore, CCN activities were also compared under nine different weather-pollution conditions
(Che et al., 2016), and $\kappa$ were determined to be ~0.7 and ~0.1 for inorganics and organics during
haze episodes and ~0.6 and ~0.2 for other episodes.
**4.3  Pearl River Delta (PRD)**
Rose et al. (Rose et al., 2010; Rose et al., 2011) explored size-resolved CCN activities in July
2006 at the Backgarden site, which is a suburban site (23.55ºN, 113.07ºE) located ~60 km
northwest of Guangzhou. Maximum activation fractions were close to 1 at medium and high
supersaturation (0.47-1.27%) and well below 1 at low supersaturation (0.068-0.27%) (Rose et al.,
2010), and particles not activated were mainly externally mixed soot with an estimated median $\kappa$
of ~0.01 (Rose et al., 2011). The average $\kappa_a$ and $\kappa_t$ were determined to be 0.34 and 0.30 over the
entire campaign; to be more specific, $\kappa_a$ and $\kappa_t$ were almost identical (~0.3) for small particles and
increased to 0.4-0.5 and ~0.33 for large particles (Rose et al., 2010). Increase in average $\kappa_a$ with
diameter was mainly due to enhanced mass fractions of inorganics for larger particles (Rose et al.,
2011). Compared to the rest of the campaign, $\kappa_a$ and $\kappa_t$ were reduced by ~30% on average during
biomass burning events (0.34 versus 0.24), when mass fractions of organics were substantially
increased; moreover, the decrease in $\kappa_t$ during biomass burning events was very substantial for
<100 nm particles but quite small for ~200 nm particles (Rose et al., 2010). It was further found
that assuming $\kappa$ to be ~0.6 for inorganics and ~0.1 for organics could approximate the observed
CCN activities over the entire campaign (Rose et al., 2011).
Size-resolved CCN activities were investigated at the Panyu site in November-December 2014
(Cai et al., 2018), and the average $\kappa_a$ were found to increase from 0.21 at 58 nm to 0.30 at 156 nm,
because mass fractions of organics, measured using AMS, decreased with particle size. The
average $\kappa$ derived from H-TDMA measurements agreed well with those derived from CCN





measurements; however, they were larger than those calculated from size-resolved chemical
compositions, and the difference between measured and calculated $\kappa$ increased with particle size
(Cai et al., 2018). This discrepancy was probably because assuming a constant $\kappa$ (0.1) may
underestimate the hygroscopicity of aerosol organics.
Aerosol CCN properties were studied at the HKUST site in May 2011 (Meng et al., 2014), and
maximum activation fractions were found to exceed 0.9 for the entire campaign, implying that the
difference between $\kappa_a$ and $\kappa_t$ should be small. CCN activities were found to increase with particle
size, with average $\kappa_a$ being determined to be 0.28 at 46 nm to 0.39 at 116 nm (Meng et al., 2014),
due to increase in volume fractions of inorganics as revealed by AMS measurements. It was further
found that the measured $\kappa_a$ could be reasonably well predicted using volume fractions of inorganics
and organics (Meng et al., 2014), and their $\kappa$ were determined to be 0.6 and 0.1.
**4.4   Other locations**
Hung et al. (Hung et al., 2014; Hung et al., 2016) measured [CCN], [CN] and aerosol number
size distribution in August 2011 at a rural site and in June 2012 at an urban site in Taiwan. The
rural site (25.89ºN, 121.57ºE) is ~15 km away from Taipei, while the urban site (25.01ºN, 121.54ºE)
is located on the campus of National Taiwan University in a metropolitan area of Taipei. At the
rural site, $\kappa_{cut}$ increased from ~0.1 at ~50 nm to ~0.35 at ~165 nm during the first period which
was significantly affected by anthropogenic emissions, while increased from ~0.04 at ~70 nm to
~0.28 at ~175 nm for the second period not significantly affected by anthropogenic emissions
(Hung et al., 2014). Overall, $\kappa_{cut}$ was larger in the first period than the second period, probably due
to the impacts of aged air masses originating from cities nearby during the first period. Compared
to the rural site, $\kappa_{cut}$ were much smaller at the urban site, increasing from ~0.021 at ~90 nm to 0.10





970 at ~250 nm (Hung et al., 2016), indicating that fresh anthropogenic aerosols tended to exhibit lower

971 hygroscopicity.

972  Shipborne size-resolved CCN activity measurements were carried out in September 2012 over

973 remote regions of the South China Sea and East China Sea (Atwood et al., 2017). Under marine

974 background conditions, the average $\kappa_a$ were determined to be 0.65±0.11 and 0.46±0.17 for the

975 accumulation and Aitken modes (Atwood et al., 2017). Compared to marine background

976 conditions, CCN activities were reduced after extensive precipitation, with average $\kappa_a$ determined

977 to be 0.54±0.14 and 0.34±0.11 for the accumulation and Aitken modes; whereas during periods

978 impacted by biomass burning, $\kappa_a$ was reduced to 0.40±0.03 for the accumulation mode but

979 increased instead to 0.56±0.25 for the Aitken mode (Atwood et al., 2017).

980  Size-resolved CCN activities were explored over north South China Sea (19º39'N to 22º43'N,

981 113º44'E to 118º12'E) in August 2018 (Cai et al., 2020), and no obvious dependence of $\kappa_a$ on

982 particle size (50-100 nm) was observed. The campaign-averaged $\kappa$ was determined to be ~0.40

983 (Cai et al., 2020), larger than these measured in the PRD region but smaller than those measured

984 over remote marine regions. This is because the air in north South China Sea was affected by both

985 continental air masses (low hygroscopicity) and marine background (high hygroscopicity).

986 **4.5 Summary**

987  Similar to H-TDMA measurements, CCN activity measurements in China were mainly carried

988 out in NCP, YRD and PRD, and almost all the measurements took place at or close to the ground

989 level. In addition, the number of CCN activity measurements is much smaller than H-TDMA

990 measurements. The limited number of field studies preclude any solid conclusions on diurnal and

991 seasonal variations of aerosol CCN activities being drawn.



Maximum activation fractions were typically found to be considerably smaller than 1 (Rose et
al., 2010; Gunthe et al., 2011; Che et al., 2017; Zhang et al., 2017), especially at low
supersaturation, and CCN-inactive particles were usually attributed to low hygroscopic primary
particles (e.g., soot) from local sources. The average $\kappa$, reported by previous studies, were
generally found to be in the range of 0.30-0.35; however, CCN activities could be significantly
reduced if measurement sites were affected by fresh urban pollution or biomass burning (Rose et
al., 2010; Gunthe et al., 2011; Zhang et al., 2014; Wu et al., 2017), due to enhanced contribution
of soot and organics. The average $\kappa$ observed at the Xinzhou site appeared to be larger than those
reported at other continental site (Zhang et al., 2017), probably because aerosols arriving at this
site were heavily aged. In addition, two studies which investigated aerosol CCN activities in the
marine boundary layer reported larger $\kappa$ values (Atwood et al., 2017; Cai et al., 2020), compared
to those at continental sites.

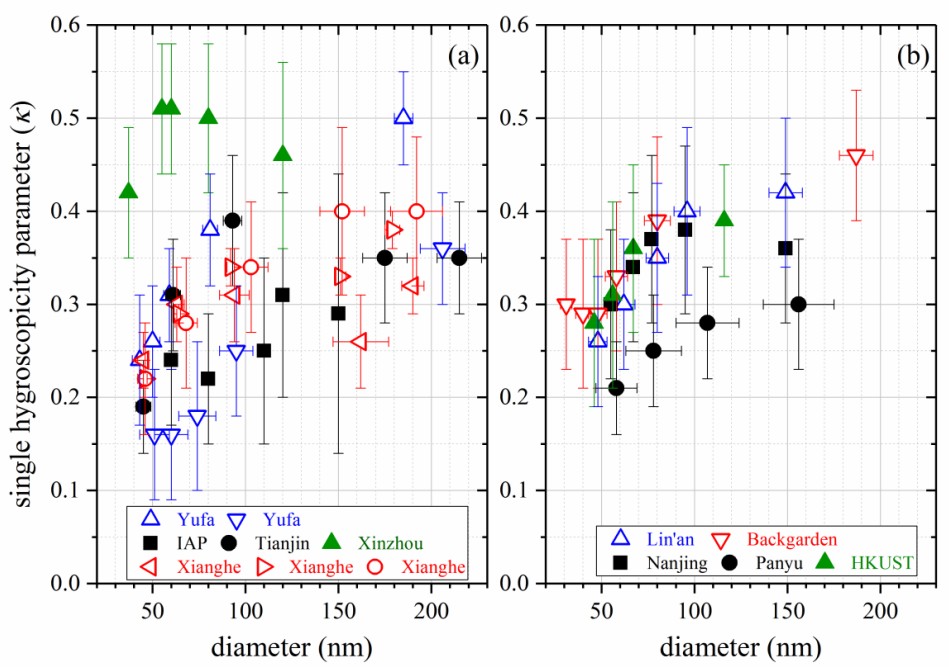






**Figure 7.** Measured $\kappa_a$ as a function of particle diameter reported by previous studies (Rose et al.,
2010; Deng et al., 2011; Gunthe et al., 2011; Deng et al., 2013; Meng et al., 2014; Zhang et al.,
2014; Che et al., 2016; Ma et al., 2016; Che et al., 2017; Ma et al., 2017; Zhang et al., 2017; Cai
et al., 2018; Tao et al., 2020) in the NCP (a) and other regions in China (b). Solid symbols represent
urban/suburban sites, and open symbols represent rural sites.

Figure 7 summarizes size dependence of $\kappa_a$ reported by CCN measurements at continental sites
in China, and measurement data related to specific cases (e.g., biomass burning events) are not
included (Rose et al., 2010; Wu et al., 2017). As shown in Figure 7, in general $\kappa_a$ increased with
particle size, as mass fractions increased with particle size for soluble inorganics and decreased for
organics. Nevertheless, no obvious dependence of $\kappa_a$ on particle size was also observed in Xinzhou
(Zhang et al., 2017) and Nanjing (Ma et al., 2017), probably because aerosol particles at these two
sites were substantially aged and thus very well internally mixed.
Several studies carried out CCN activity closure analysis. Some studies suggested that the
measured $\kappa$ could be well quantitatively explained by aerosol composition (Rose et al., 2010;
Gunthe et al., 2011; Wu et al., 2017), while other studies showed that $\kappa$ estimated using aerosol
composition were either larger (Zhang et al., 2017) or smaller than measured values (Zhang et al.,
2017; Cai et al., 2018). In addition, a few studies investigated aerosol hygroscopic growth and
CCN activities concurrently, and both consistence (Wu et al., 2017; Zhang et al., 2017; Cai et al.,
2018) and discrepancies (Ma et al., 2016; Zhang et al., 2016b) were reported.
**5 Perspectives**
In the last 10-20 years a number of field measurements of hygroscopic properties and CCN
activities of tropospheric aerosols have been carried out in China, and summaries of measured



hygroscopic properties and CCN activities are provided in Sections 3.5 and 4.5. As shown in

Sections 3 and 4, these studies have significantly improved our knowledge of tropospheric aerosol

hygroscopicity in China and provided valuable data to better understand the roles aerosols play in

heterogeneous and multiphase chemistry, as well as direct and indirect radiative forcing. However,

large knowledge gaps still exist for aerosol hygroscopicity in China, as described below, and future

research directions are also discussed.

**Data availability:** In Tables S1-S5 we attempt to compile measurement data reported by

previous studies under a consistent framework in order to enhance their accessibility and usability.

However, important data are not always available from every study published; for example, several

studies presented their main results graphically. It is recommended that in future data in the

numerical form (H-TDMA measurements: including but not limited to diameter, RH, and GF

and/or $\kappa$; CCN activity measurements: including but not limited to supersaturation, activation

diameter and $\kappa$) should be provided.

**Geographical coverages:** As shown in Sections 3-4, almost all the measurements of

hygroscopic properties and CCN activities in China were carried out in east regions (e.g., NCP,

YRD and PRD) heavily affected by anthropogenic emissions. Therefore, it will be very desirable

in future to carry out these measurements in other regions; measurements in areas far from by

human activities will be especially important, as they will provide information on aerosol

hygroscopicity in the pristine troposphere.

**Vertical distribution:** Most of previous aerosol hygroscopicity measurements in China were

only carried out at or close to the ground level. However, both aerosol composition and RH, and

as a result aerosol hygroscopic growth and CCN activation, will vary with altitude. For example,

aircraft-based measurements of aerosol size distribution and composition indicated that single





hygroscopicity parameters would increase significantly with altitude (Liu et al., 2020), and it was

revealed from remote sensing that aerosol hygroscopicity at the upper boundary level was different

from that at the ground level (Tan et al., 2020). Therefore, in-situ measurements of vertical profiles

of aerosol composition and hygroscopicity on different platforms (e.g., towers, airships, aircrafts,

and etc.) will be very valuable; in addition, remote sensing may be very useful for retrieving

vertical profiles of aerosol hygroscopicity, as demonstrated by a very recent study (Tan et al., 2020).

**Long-term measurements:** Both aerosol concentration and composition have undergone (and

very likely will undergo) significant changes in China; however, most aerosol hygroscopicity

measurements were carried out for 1-2 months during specific field campaigns. Long-term

measurements of aerosol hygroscopicity will be very important to understand seasonal and annual

variations of aerosol hygroscopicity and the implications for visibility, atmospheric chemistry and

climate change.

**Hygroscopicity of large particles:** Tables S1-S4 reveal that the maximum aerosol diameter

examined in hygroscopic growth studies was 350 nm, which is the upper limit of dry aerosol size

for most of H-TDMA instruments (Tang et al., 2019). As particles larger than 350 nm can

contribute substantially to aerosol surface area and volume (or mass) concentrations,

hygroscopicity of these particles will be very important and should be measured in future, and this

requires technical improvements of H-TDMA. On the other hand, hygroscopicity of >350 nm

particles may not be very important for CCN activation, as these particles can be easily activated

due to their large diameters.

**RH dependence:** Most H-TDMA measurements were carried out at a single RH (usually

~90%), and a few studies which measured GF as a function of RH suggested that a constant $\kappa$

failed to describe hygroscopic growth at different RH. In addition, due to lack of measurement



data at different RH, it is not clear how well widely-used aerosol thermodynamic models can
simulate ALWC at ambient RH. Therefore, measurements of aerosol hygroscopicity at different
RH are certainly warranted.
**The effect of aerosol organics:** As discussed in Section 3, several studies (Liu et al., 2014;
Wu et al., 2016; Cai et al., 2018; Hong et al., 2018; Li et al., 2019b; Jin et al., 2020) suggested that
organics contributed substantially to aerosol water uptake, while some studies also indicated that
the contribution of aerosol organics to ALWC was rather minor. Therefore, aerosol hygroscopicity
closure analysis, with concurrent measurements of aerosol composition and hygroscopicity, is
recommended for future, in order to further understand the effects of aerosol organics on ALWC
and CCN activation; in addition, relevant factors which need consideration include the dependence
of hygroscopicity on composition of aerosol organics (e.g., O/C ratios) and the effects of aerosol
organics on surface tension, phase separation effects, and etc.

**Data Availability.** This is a review paper, and all the data used come from cited literature. In
addition, the data we have compiled can be found in the supplement.
**Author contribution.** Mingjin Tang conceived this work; Chao Peng and Mingjin Tang wrote the
manuscript with substantial input from Yu Wang, Zhijun Wu and Lanxiadi Chen; all the authors
revised the manuscript and approved its submission.
**Conflict of Interest.** The authors declare no conflict of interest.
**Acknowledgment.** We would like to thank participants in the fifth International Workshop on
Heterogeneous Kinetics Related to Atmospheric Aerosols for discussion.
**Financial support.** This work was funded by National Natural Science Foundation of China
(91744204 and 91844301), State Key Laboratory of Organic Geochemistry (SKLOG2016-A05),



State Key Laboratory of Loess and Quaternary Geology (SKLLQG1921), Department of Science
and Technology of Guangdong Province (2017GC010501), and Guangdong Foundation for
Program of Science and Technology Research (2017B030314057 and 2019B121205006). Mingjin
Tang would like to thank the CAS Pioneer Hundred Talents program for providing a starting grant.

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
