# Peer review of "Tropospheric aerosol hygroscopicity measurements in China"

_Atmospheric Chemistry and Physics, 2020_

## Author Comment (AC1) · 21 Jun 2020

26 May 2020

Dear Professor Jingkun Jiang,

Thank you very much for handling our manuscript (acp-2020-386). We have carefully considered comments provided by you and ref #1, and revised the manuscript accordingly. Changes to the original manuscript are highlighted in red in the revised manuscript, and in this letter we summarize in brief our response and changes we have made. To facilitate communications between you, referees and us, and to foster open discussions and peer-review, we will also post this letter online as an "author comment" after our manuscript is published online as a discussion paper.

Ref #1 pointed out that some original research papers were not cited in our original manuscript. We are aware of these work; we did not cite them in our original manuscript because some review articles we cited include these information and we wanted to make our manuscript concise. As recommended by ref #1 as well as you, in the revised manuscript we have added two paragraphs in Section 2.1 and another two paragraphs in Section 2.2 (page 6-9, highlighted in red); in addition, we have also cited a number of representative papers in which original research is presented, in order to better balance the literature coverage.

We agree with ref #1 that discussion of previous work reviewed in our manuscript can be further enhanced, and we especially agree with your comment "Though the topic of this manuscript focuses on China, these discussions can be from a global perspective". As revision like this may take quite a while, we prefer to substantially improve our manuscript in this and other aspects after full reviews of our manuscripts are available, upon your permission. Therefore, could you please consider having our manuscript accepted for open discussion without revision in this aspect?

We would like to thank you and all the three referees for the precious time spent on reviewing our manuscript.

Sincerely,

Dr. Mingjin Tang
Guangzhou Institute of Geochemistry
Chinese Academy of Sciences

---

## Referee Comment (RC1) · Anonymous Referee #2 · 19 Jul 2020

Peng et al. overview aerosol hygroscopicity data in China in terms of hygroscopic parameter and CCN activity. This review paper seems to simply list the results reported by earlier work without much interpretation based on atmospheric chemistry. For example, the hygroscopic parameters varied with particle size, chemical composition (primary vs secondary species), seasonal effect, and so on. This paper lacks the discussions on how the variable hygroscopic parameters are related to many factors mentioned above. I recommend the authors to provide more underlying mechanisms about the hygroscopicity characteristic in China and otherwise the paper would be just collection of the results. This current manuscript is not ready for publication and requires major revisions.

The introduction needs improvements. The authors describe their motivation to report the current study with fairly rational information. However, it still lacks why one needs to review aerosol hygroscopicity in China now under what circumstances. Elaborating such points would put this study in better context.

The title would be misleading. It seems to me that this study focuses on aerosol hygroscopicity in China, but not its measurements.

I am aware of the recent review paper on aerosol hygroscopicity (Tang et al., 2019). It is not convincing at all if the authors do not provide any elaboration on how the current study is distinguishable by Tang et al.

Line 97: I do not agree that the single particle studies are too limited for the overall aerosol hygroscopicity. This type of lab studies has proven powerful to establish aerosol thermodynamic models that can provide useful information on the overall aerosol hygroscopicity. It is understood that this manuscript focuses on field measurements of ambient aerosol hygroscopicity, but it needs to rephrase the argument.

Line 200: Can you explain why is there no obvious difference in aerosol hygroscopicity between summer and winter?

In Figure 1, the hygroscopic parameter was always highest in summer and lowest in winter. Photochemical processes and secondary products play a role in this trend. Can you also explain the seasonal difference in the hygroscopic parameter in terms of chemical composition difference? For instance, what is the major inorganic and organic species between two seasons and O/C ratios?

In Figure 2, it is related to the questions above. The hygroscopic parameter increased with particle size, which was attributed to enhanced contribution of secondary species. Can you provide more information on how chemical composition of PM changes with particle size increase?

Lines 319-322: How did the size-resolved mass fractions of secondary inorganic species lead to slight decrease in the hygroscopic parameter in winter as particle size increases?

Line 856: Why there was no size dependence of the hygroscopic parameters of activated particles despite the large values?

Lines 952-955: The logic is not clear. The authors mentioned that If measurement sites were affected by primary emissions, CCN activities could be reduced. However, if so, the averaged hygroscopic parameters can be reduced too. I imagine that CCN activities would be essentially attributable to the hygroscopic parameters. How can the contribution of soot and organics reduce CCN activities while the hygroscopic parameters remain high?

Lines 979-980: What are potential reasons for the consistence and discrepancies?

Minor points: Line 217: A typo of "exntensive". Line 544: One possible reason for what? Line 619: A typo of "he". Line 760: You mean "exceptions"? Line 823: Please add "respectively". Were the hygroscopic parameters for organics assumed to be 0?

---

## Referee Comment (RC2) · Anonymous Referee #3 · 22 Aug 2020

The review by Peng et al. is an ambitious study in trying to summarize aerosol hygroscopicity measurements in China. The authors efforts are commendable and will certainly guide the future research efforts (at least in China and beyond). The review is well written and easy to follow, so should be acceptable for publication after addressing the comments.

One major comment is arising from the author's efforts to make a fair summary of all the measurements, but without connecting observations with processes and/or sources. As such, reading the large portions of text becomes boring, because it only mentions facts (easily found in individual papers by concerned readers) without linking or extending scientific knowledge. The review is not only meant to provide a summary of observations (that would be rather a report, not scientific study), but most importantly

to critically analyse available knowledge and subsequently to identify scientific knowledge gaps. CCN part of the review is written much better, but HTDMA part is lacking interpretation on every page or even more often. Few good and bad examples were noted, but the authors should read their text carefully to recognise the rest.

The second major comment relates to uncertainty analysis and even more importantly taking into account that uncertainty when interpreting the results of various studies. "Smaller" or "larger" is irrelevant on absolute scale, it is only important when the differences are outside the uncertainty range of GF or kappa. When the differences are within the uncertainty range it should be stated accordingly. Therefore, it advised to carefully use the words "different", similar" which carry very little scientific significance.

The abstract is currently a very formal structural summary when instead it should be a scientific one, highlighting identified knowledge gaps (perhaps, limiting to the most important ones). It should give a flavour what was uncovered by the review and engage the reader.

Minor comments

Line 164. typo in 0.25, same in next instance.

Figure 1 contains no error bars.

Line 345. It is important for the review paper to give an in depth explanation of the observed phenomenon, not just acknowledge that differences were observed. Diurnal patterns must come from either dynamics of BL, photochemistry or sources, or interplay of the three.

Line 413. Same comment about summarizing observations without linking to processes and sources. Observed bimodality typically means different sources like traffic and secondary aerosol formation.

Line 417. It should be specifically reworded: "It was found that secondary inorganic aerosol species increased hygroscopic growth of accumulation mode while organics

were decreasing hygroscopicity of the Aitken mode".

Line 429. ...suggesting that ISORPIA-II was not capable to reproduce ALWC at low RH.

Line 437. Same comment on observations versus processes.

Line 488. Again missing comment as to what bimodality and increasing kappa means.

Line 588. Good example of trying to explain the observations and link to composition and sources, not just documenting them.

Line 596. Good example

Line 673. ...or internal/external mixtures of organic and inorganic compounds.

Line 697. Can the reason be discerned? Well mixed aged aerosol removing differences of various sources of origin?

Line 702. organic matter, not materials

Line 726. if evident state the number of higher RH. Was it evident at 90%?

Line 758. ...almost all of the...

Line 766. Not even in summary there is interpretation what multimodal hygroscopicity means in terms of processes and sources.

Line 775. How different? Opposite?

Line 781. However, Meier et al. (2009) found that primary particles smaller than about 50nm in diameter exhibited decreasing hygroscopicity. If I interpreted correctly.

Line 789. The results should be interpreted in terms of processes and sources.

Line 796. That was already stated numerous times, no need to repeat. The paragraph should start with underlying reasons.

Line 815. If kappa is considered a robust method, it does not matter at which RH GF was measured at, because lower RH would result in lower GF and kappa should be the same. If it was not the same, then that should be highlighted by proper comparison and stated clearly, because that is very important. Not all species exhibit hysteresis and even fewer when internally mixed.

Line 824. NaCl has the highest deliquescence of 75% among the relevant atmospheric species, so the statement should state that no kappa(HTDMA) should be derived below 75-80%. The following Figure is manifesting that, but needs error bars added to data points.

Figure 5. Uncertainty of the calculated kappa is clearly above 10% based on very basic considerations. If one considers size uncertainty of two independent DMA at 10% each and RH measurement which is inherently drifting during HTDMA operation, one would get ∼17% total uncertainty. Therefore, no one can objectively claim kappa differences of ∼10%, because those will be within the overlapping error bars.

Line 877. was lower, not became lower. There is more to it. Calculated (chemical) kappa is relying on compound specific kappa values, which have uncertainty and without even mentioning rather arbitrary kappa of organic matter.

Line 883. . . .while the increase in aerosol hygroscopicity was much smaller due to the change in chemical composition.

Line 897. Was that outside uncertainty range?

Line 1023. . . .and both consistencies.... and discrepancies were reported

Line 1033. . . .research directions can be proposed.

Line 1042. . . .in eastern regions

Line 1046. . . .hygroscopicity in the cleaner troposphere. "Pristine" can only possibly apply to remote oceanic regions or Antarctica. Not even Arctic is pristine.

Line 1069. ...can be easily activated at the lowest supersaturation due to their size.

Line 1074. It should be stated that kappa(HTDMA) derivation should be limited to RH above 75-80% due to reasons discussed.

---

## Author Comment (AC2) · 2 Sep 2020

Comments by referees are in blue.
Our replies are in black.
Changes to the manuscript are highlighted in red both here and in the revised manuscript.

**Reply to referee #2**

Peng et al. overview aerosol hygroscopicity data in China in terms of hygroscopic parameter and CCN activity. This review paper seems to simply list the results reported by earlier work without much interpretation based on atmospheric chemistry. For example, the hygroscopic parameters varied with particle size, chemical composition (primary vs secondary species), seasonal effect, and so on. This paper lacks the discussions on how the variable hygroscopic parameters are related to many factors mentioned above. I recommend the authors to provide more underlying mechanisms about the hygroscopicity characteristic in China and otherwise the paper would be just collection of the results. This current manuscript is not ready for publication and requires major revisions.

**Reply:** We would like to thank ref #2 for reviewing our manuscript. We agree that in our review we should further discuss factors related to aerosol hygroscopicity. In the revised manuscript, we have made large efforts to explain/interpret underlying mechanisms for aerosol hygroscopicity in China. For more information about changes we have made, we kindly refer ref #2 to the revised manuscript as well as our responses to specific comments raised by the two referees.

The introduction needs improvements. The authors describe their motivation to report the current study with fairly rational information. However, it still lacks why one needs to review aerosol hygroscopicity in China now under what circumstances. Elaborating such points would put this study in better context.

**Reply:** In response to this comment, in the revised manuscript we have added a few sentences in the fourth paragraph in the introduction Section (page 5) to explain why we would like to write this review paper: "In the last few decades, a number of field studies have investigated tropospheric aerosol hygroscopicity in China. However, a general overview of spatial and temporal variation of aerosol hygroscopicity in China is yet to be provided, and the dependence of aerosol hygroscopicity on aerosol composition, mixing state, and etc. has not been fully elucidated. In this paper we provide a comprehensive review of hygroscopic properties of ambient aerosols measured using H-TDMA in China; in addition, CCN activities of tropospheric aerosols measured in China are also reviewed and discussed."

The title would be misleading. It seems to me that this study focuses on aerosol hygroscopicity in China, but not its measurements.

**Reply:** The referee is right. We have changed the title to "Tropospheric aerosol hygroscopicity in China".

I am aware of the recent review paper on aerosol hygroscopicity (Tang et al., 2019). It is not convincing at all if the authors do not provide any elaboration on how the current study is distinguishable by Tang et al.

**Reply:** The recent review paper by Tang et al. (2019) is focused on the experimental techniques for aerosol hygroscopicity measurements. In the revised manuscript (page 5), we have expanded the sentence to further clarify the difference between our current review and the review paper by Tang et al. (2019): "In addition, a recent paper (Tang et al. 2019) has reviewed aerosol hygroscopicity measurement techniques, but it only discussed several exemplary studies to

illustrate how specific techniques can help us better understand tropospheric aerosol hygroscopicity."

Line 97: I do not agree that the single particle studies are too limited for the overall aerosol hygroscopicity. This type of lab studies has proven powerful to establish aerosol thermodynamic models that can provide useful information on the overall aerosol hygroscopicity. It is understood that this manuscript focuses on field measurements of ambient aerosol hygroscopicity, but it needs to rephrase the argument.

**Reply:** The referee is right. In the revised manuscript (page 5) we have rephrased our argument to avoid misleading implications: "Single particles techniques (Krieger et al., 2012; Li et al., 2016) have provided physiochemical data which are very valuable to test aerosol thermodynamic models, largely helping us better understand tropospheric aerosol hygroscopicity. However, as numbers of particles examined in single particle studies are very limited, these studies usually do not provide direct information of overall aerosol hygroscopicity in the ambient air and thus are not discussed herein."

Line 200: Can you explain why is there no obvious difference in aerosol hygroscopicity between summer and winter?

**Reply:** As suggested, we have added one sentence in the revised manuscript (page 12) to explain the observation: "In addition, no obvious difference in aerosol hygroscopicity was found between summer and winter, because constantly high mass fractions (~50% wt) of carbonaceous materials (nearly hydrophobic or less hygroscopic), related to extensive usage of fossil fuel, were observed in both seasons for submicrometer particles (Massling et al., 2009)."

In Figure 1, the hygroscopic parameter was always highest in summer and lowest in winter. Photochemical processes and secondary products play a role in this trend. Can you also explain the seasonal difference in the hygroscopic parameter in terms of chemical composition difference? For instance, what is the major inorganic and organic species between two seasons and O/C ratios?

**Reply:** We agree with the referee that photochemical processes and enhanced formation of secondary species play a role in enhanced aerosol hygroscopicity in summer, compared to winter; however, the original paper provided no information on seasonal variation in aerosol composition. To address this comment, in the revised manuscript (page 15) we have added one sentence to explain the possible reason for the observed seasonal variation in aerosol hygroscopicity: "The difference in aerosol hygroscopicity between summer and winter may be caused by enhanced photochemical processes in the summer and as a result increased fractions of secondary species."

In Figure 2, it is related to the questions above. The hygroscopic parameter increased with particle size, which was attributed to enhanced contribution of secondary species. Can you provide more information on how chemical composition of PM changes with particle size increase?

**Reply:** In response to this comment, we have added one sentence in the revised manuscript (page 16-17) to explain the variation of aerosol hygroscopicity with particle size: "Figure 2 shows that aerosol hygroscopicity increased with particle size and pollution level (Wang et al., 2017d), because mass fractions of hydrophilic species, such as sulfate, nitrate and oxidized organics increased with particle size, especially under highly polluted conditions."

Lines 319-322: How did the size-resolved mass fractions of secondary inorganic species lead to slight decrease in the hygroscopic parameter in winter as particle size increases?

**Reply:** From the size-resolved H-TDMA and HR-TOF-AMS measurements, the average $\kappa$ slightly decreased from 0.271 at 110 nm to 0.260 at 200 nm, while the mass fractions of secondary inorganic species decreased from 61.8% at 110 nm to 59.3% at 200 nm. In the revised manuscript (page 18) we have made the following changes to provide further explanation: "The average $\kappa$

increased from 0.158 at 40 nm to 0.271 at 110 nm in winter, and further increase in particle size (to 200 nm) led to slight decrease in $\kappa$, because mass fractions of secondary inorganic species decreased slightly from 61.8% at 110 nm to 59.3% at 200 nm (Fan et al., 2020). For comparison, the average $\kappa$ increased with particles size in summer, from 0.211 at 40 nm to 0.267 at 200 nm, as mass fractions of secondary inorganic species increased from 56.7% at 80 nm to 63.0% at 200 nm (Wang et al., 2019b; Fan et al., 2020)."

Line 856: Why there was no size dependence of the hygroscopic parameters of activated particles despite the large values?

**Reply:** Aerosol particles in Xinzhou was highly aged and well internally mixed, and the variation of chemical compositions with particle size was negligible; therefore, $\kappa$ were larger compared to other sites in the NCP and showed no obvious size dependence. We have made following changes in the revised manuscript (page 45) to provide further explanation: "This is because aerosols observed at this site were highly aged and well internally mixed after undergoing regional transport for a long time, and thus the variation of chemical compositions with particle size was negligible."

Lines 952-955: The logic is not clear. The authors mentioned that If measurement sites were affected by primary emissions, CCN activities could be reduced. However, if so, the averaged hygroscopic parameters can be reduced too. I imagine that CCN activities would be essentially attributable to the hygroscopic parameters. How can the contribution of soot and organics reduce CCN activities while the hygroscopic parameters remain high?

**Reply:** Indeed if significantly affected by primary emissions, both hygroscopic properties (RH <100%) and CCN activities (RH>100%) should be reduced, as suggested by several studies (Rose et al., 2010; Gunthe et al., 2011; Zhang et al., 2014; Wu et al., 2017) mentioned in this paragraph. The work by Zhang et al. (2017) found high aerosol hygroscopicity, but aerosols investigated by this work were heavily aged, instead of being affected by primary emissions. In order to reduce confusion, in the revised manuscript (page 49) we have made the following changes: "…due to enhanced contribution of soot and organics. We note that a few recent studies (Atwood et al., 2017; Zhang et al., 2017; Cai et al., 2020) also reported higher aerosol hygroscopicity. For example, the average $\kappa$ observed at the Xinzhou site appeared to be larger than those reported at other continental site (Zhang et al., 2017)…"

Lines 979-980: What are potential reasons for the consistence and discrepancies?

**Reply:** Ideally, they should be consistent, while discrepancies were not usual due to several reasons, including solution non-ideality of aerosol droplets, limited solubility of some components contained by aerosol particles, surface tension effects, and etc. We have added one sentence in the revised manuscript (page 51) to provide further explanation: "The discrepancies could be caused by several factors (Petters and Kreidenweis, 2008; Wex et al., 2009; Petters and Kreidenweis, 2013; Liu et al., 2018), such as solution non-ideality of aerosol droplets, limited solubility of some components contained by aerosol particles, surface tension effects, and etc."

Minor points:

Line 217: A typo of "exntensive".

**Reply:** Corrected in the revised manuscript (page 13).

Line 544: One possible reason for what?

**Reply:** In the revised manuscript (page 30) we have provided further information to clarify it: "One possible reason for difference in aerosol hygroscopicity observed in the two periods at the same site was that in April-May 2014 organic species made a large contribution to submicrometer aerosols…"

Line 619: A typo of "he".

    **Reply:** Corrected in the revised manuscript (page 33).

Line 760: You mean "exceptions"?

    **Reply:** That is right, and we have corrected it in the revised manuscript (page 40).

Line 823: Please add "respectively". Were the hygroscopic parameters for organics assumed to be 0?

    **Reply:** The hygroscopicity parameter was indeed assumed to be 0 for organics. We have rephrased this sentence in the revised manuscript (page 43) to be clearer: "…if $\kappa$ were assumed to be 0.53 for inorganics and 0 for organics, respectively."

**Reference**

Atwood, S. A., Reid, J. S., Kreidenweis, S. M., Blake, D. R., Jonsson, H. H., Lagrosas, N. D., Xian, P., Reid, E. A., Sessions, W. R., and Simpas, J. B.: Size-resolved aerosol and cloud condensation nuclei (CCN) properties in the remote marine South China Sea - Part 1: Observations and source classification, Atmospheric Chemistry and Physics, 17, 1105-1123, 2017.

Cai, M., Liang, B., Sun, Q., Zhou, S., Yuan, B., Shao, M., Tan, H., and Zhao, J.: Effects of continental emissions on Cloud Condensation Nuclei (CCN) activity in northern South China Sea during summertime 2018, Atmospheric Chemistry and Physics Discussions, 2020, 1-43, 2020.

Krieger, U. K., Marcolli, C., and Reid, J. P.: Exploring the complexity of aerosol particle properties and processes using single particle techniques, Chemical Society Reviews, 41, 6631-6662, 2012.

Li, W., Shao, L., Zhang, D., Ro, C.-U., Hu, M., Bi, X., Geng, H., Matsuki, A., Niu, H., and Chen, J.: A review of single aerosol particle studies in the atmosphere of East Asia: morphology, mixing state, source, and heterogeneous reactions, Journal of Cleaner Production, 112, 1330-1349, 2016.

Liu, P., Song, M., Zhao, T., Gunthe, S. S., Ham, S., He, Y., Qin, Y. M., Gong, Z., Amorim, J. C., Bertram, A. K., and Martin, S. T.: Resolving the mechanisms of hygroscopic growth and cloud condensation nuclei activity for organic particulate matter, Nature Communications, 9, 2018.

Massling, A., Stock, M., Wehner, B., Wu, Z. J., Hu, M., Brueggemann, E., Gnauk, T., Herrmann, H., and Wiedensohler, A.: Size segregated water uptake of the urban submicrometer aerosol in Beijing, Atmospheric Environment, 43, 1578-1589, 2009.

Petters, M. D., and Kreidenweis, S. M.: A single parameter representation of hygroscopic growth and cloud condensation nucleus activity - Part 2: Including solubility, Atmospheric Chemistry and Physics, 8, 6273-6279, 2008.

Petters, M. D., and Kreidenweis, S. M.: A single parameter representation of hygroscopic growth and cloud condensation nucleus activity - Part 3: Including surfactant partitioning, Atmospheric Chemistry and Physics, 13, 1081-1091, 2013.

Tang, M., Chan, C. K., Li, Y. J., Su, H., Ma, Q., Wu, Z., Zhang, G., Wang, Z., Ge, M., Hu, M., He, H., and Wang, X.: A review of experimental techniques for aerosol hygroscopicity studies, Atmospheric Chemistry and Physics, 19, 12631-12686, 2019.

Wang, Y., Zhang, F., Li, Z., Tan, H., Xu, H., Ren, J., Zhao, J., Du, W., and Sun, Y.: Enhanced hydrophobicity and volatility of submicron aerosols under severe emission control conditions in Beijing, Atmospheric Chemistry and Physics, 17, 5239-5251, 2017d.

Wex, H., Petters, M., Carrico, C., Hallbauer, E., Massling, A., McMeeking, G., Poulain, L., Wu, Z., Kreidenweis, S., and Stratmann, F.: Towards closing the gap between hygroscopic growth and activation for secondary organic aerosol: Part 1–Evidence from measurements, Atmospheric Chemistry and Physics, 9, 3987-3997, 2009.

Zhang, F., Wang, Y., Peng, J., Ren, J., Collins, D., Zhang, R., Sun, Y., Yang, X., and Li, Z.: Uncertainty in Predicting CCN Activity of Aged and Primary Aerosols, Journal of Geophysical Research-Atmospheres, 122, 11723-11736, 2017.

---

## Author Comment (AC3)

Comments by referees are in blue.
Our replies are in black.
Changes to the manuscript are highlighted in red both here and in the revised manuscript.

**Reply to referee #3**

The review by Peng et al. is an ambitious study in trying to summarize aerosol hygroscopicity measurements in China. The authors efforts are commendable and will certainly guide the future research efforts (at least in China and beyond). The review is well written and easy to follow, so should be acceptable for publication after addressing the comments.

**Reply:** We would like to thank ref #3 for reviewing our manuscript and recommending it for publication after revision. His/her comments, which helped us largely improve our manuscript, have been carefully addressed in our revision, as detailed below.

One major comment is arising from the author's efforts to make a fair summary of all the measurements, but without connecting observations with processes and/or sources. As such, reading the large portions of text becomes boring, because it only mentions facts (easily found in individual papers by concerned readers) without linking or extending scientific knowledge. The review is not only meant to provide a summary of observations (that would be rather a report, not scientific study), but most importantly to critically analyse available knowledge and subsequently to identify scientific knowledge gaps. CCN part of the review is written much better, but HTDMA part is lacking interpretation on every page or even more often. Few good and bad examples were noted, but the authors should read their text carefully to recognise the rest.

**Reply:** This is a very good point. In the revised manuscript, we have made large efforts to link reported aerosol hygroscopicity with aerosol composition, processes and sources. As detailed below, we have addressed specific comments raised by the ref #3 and revised the manuscript accordingly. Furthermore, we have additionally provided explanations/interpretations for other observations reported in previous work, and changes can be found in the revised manuscript (e.g., page 12, page 17, page 19, page 24, page 27, page 32-33).

The second major comment relates to uncertainty analysis and even more importantly taking into account that uncertainty when interpreting the results of various studies. "Smaller" or "larger" is irrelevant on absolute scale, it is only important when the differences are outside the uncertainty range of GF or kappa. When the differences are within the uncertainty range it should be stated accordingly. Therefore, it advised to carefully use the words "different", similar" which carry very little scientific significance.

**Reply:** We understand and completely agree with this concern, and the following changes have been made in the revised manuscript: 1) we have included error bars for the data shown in figures and included uncertainties for the numerical numbers in the main text, when data uncertainties are available in the original papers; 2) as suggested, when we compare measurement data reported, we are cautious when words such as "different", "similar", and etc. are used.

However, it is not always possible to be statistically rigorous when we compare measurements reported, especially for a review paper. In fact, statistically rigorous comparisons are very rare in the original work covered in this review.

The abstract is currently a very formal structural summary when instead it should be a scientific one, highlighting identified knowledge gaps (perhaps, limiting to the most important ones). It should give a flavour what was uncovered by the review and engage the reader.

**Reply:** We agree with the referee that our abstract is a structural summary, instead of being a scientific one; indeed it will be very nice if we can highlight some major findings and knowledge

gaps. However, for such a big topic in which many studies have been conducted, we find it very difficult to summarize major findings and knowledge gaps in a few sentences in the abstract. Therefore, we would like to use a structural summary to tell readers what we have done in this review paper, and interested readers can refer to the manuscript and/or individual sections for further information.

Minor comments

Line 164. typo in 0.25, same in next instance.

**Reply:** Both cases have been corrected in the revised manuscript (page 8).

Figure 1 contains no error bars.

**Reply:** We have added error bars to Figure 1 in the revised manuscript (page 14).

Line 345. It is important for the review paper to give an in depth explanation of the observed phenomenon, not just acknowledge that differences were observed. Diurnal patterns must come from either dynamics of BL, photochemistry or sources, or interplay of the three.

**Reply:** We agree that it will be very desirable to explain the observed diurnal variation in aerosol hygroscopicity. However, the diurnal variation in aerosol hygroscopicity, reported by Wang et al. (2019b), was quite complex, and the explanation they provided in the original paper was even much more complex. It is very difficult for a review paper to summarize their major findings and explanations using a few sentences. As a result, in our review paper we only mention this aspect in brief.

Line 413. Same comment about summarizing observations without linking to processes and sources. Observed bimodality typically means different sources like traffic and secondary aerosol formation.

**Reply:** As suggested, in the revised manuscript (page 21) we have expanded these sentences to provide additional explanation to the observation: "Bimodal hygroscopicity distribution, with a dominant more-hygroscopic mode and a smaller nearly-hydrophobic mode, was observed over the whole period, and the average $\kappa$, derived from GF measured at 90% RH, increased from 0.250 at 50 nm to 0.340 at 250 nm, as number fractions of aerosol particles in the more-hygroscopic mode increased with particle size (from 68% for 50 nm to 85% for 250 nm) (Liu et al., 2011). Compared to the nighttime, both the average $\kappa$ and number fractions of particles in the more-hygroscopic mode were larger during the daytime (Liu et al., 2011), because photochemical processes during the daytime led to enhanced formation of secondary species in aerosol particles and thus increase in their hygroscopicity."

Line 417. It should be specifically reworded: "It was found that secondary inorganic aerosol species increased hygroscopic growth of accumulation mode while organics were decreasing hygroscopicity of the Aitken mode".

**Reply:** As suggested, in the revised manuscript (page 21) we have rephrased this sentence in order to be more specific: "It was found that secondary inorganic species increased hygroscopicity of the accumulation mode while organics decreased hygroscopicity of the Aitken mode (Liu et al., 2014)."

Line 429. ...suggesting that ISORPIA-II was not capable to reproduce ALWC at low RH.

**Reply:** As suggested, in the revised manuscript (page 21) we have made the following change: "…but was much larger compared to the second method when ambient RH was <60% (Bian et al., 2014), suggesting that ISORROPIA-II was not capable to predict ALWC at low RH."

Line 437. Same comment on observations versus processes.

**Reply:** The following explanation has been provided in the revised manuscript (page 22) to explain the observation: "It was also found that for the accumulation mode, $\kappa$ were larger in the

nighttime than the daytime (Ding et al., 2019), as increase in RH during the nighttime led to enhanced formation of sulfate and nitrate from aqueous oxidations of $SO_2$ and heterogeneous hydrolysis of $N_2O_5$ (Wang et al., 2017a)."

Line 488. Again missing comment as to what bimodality and increasing kappa means.

**Reply:** In the revised manuscript (page 25) we have provided further explanation: "Bimodal aerosol hygroscopicity distribution was always observed (Wang et al., 2017c), indicating that aerosol particles were externally mixed. As larger particles contain higher mass fractions of secondary inorganic species, the average $\kappa$ were found to increase with particle size, from 0.240 at 30 nm to 0.320 at 250 nm."

Line 588. Good example of trying to explain the observations and link to composition and sources, not just documenting them.

**Reply:** We would like to thank the referee for his/her kind and positive comment.

Line 596. Good example

**Reply:** We would like to thank the referee for his/her kind and positive comment.

Line 673. ...or internal/external mixtures of organic and inorganic compounds.

**Reply:** In the revised manuscript (page 34) we have changed the sentence to "…may underestimate hygroscopicity of aerosol organics or mixed inorganic/organic aerosols."

Line 697. Can the reason be discerned? Well mixed aged aerosol removing differences of various sources of origin?

**Reply:** The referee is right. In the revised manuscript (page 35) we have expanded this sentence to provide some explanation: "Since aerosol particles arriving at this site were heavily aged and well internally mixed, no obvious dependence of average $\kappa$ on particle size was found."

Line 702. organic matter, not materials

**Reply:** Corrected in the revised manuscript (page 35)

Line 726. if evident state the number of higher RH. Was it evident at 90%?

**Reply:** Here higher RH means 80% and 85%. In the revised manuscript (page 36) we have modified this sentence to be more specific: "…bimodal growth behavior appeared at ~75% RH except 40 nm particles and became more evident at higher RH (80% and 85%) (Wu et al., 2018a)."

Line 758. ...almost all of the...

**Reply:** Corrected in the revised manuscript (page 37).

Line 766. Not even in summary there is interpretation what multimodal hygroscopicity means in terms of processes and sources.

**Reply:** In the revised manuscript (page 38) we have made following changes to provide further interpretation: "Bimodal or trimodal hygroscopicity distributions suggested that aerosol particles under investigation were externally mixed. Quasi-unimodal hygroscopicity distributions existed but were quite sparse (Chen et al., 2016; Wang et al., 2016; Wang et al., 2017d; Zhang et al., 2017; Wang et al., 2018b), implying that these aerosols were nearly internally mixed."

Line 775. How different? Opposite?

**Reply:** The referee is right. In the revised manuscript (page 38) we have made the following changes accordingly: "…though opposite results were also reported in several studies"

Line 781. However, Meier et al. (2009) found that primary particles smaller than about 50nm in diameter exhibited decreasing hygroscopicity. If I interpreted correctly.

**Reply:** The referee is right. In the revised manuscript (page 39) we have made the following changes: "However, different results were also reported, especially for particles at or below 50 nm (Achtert et al., 2009; Meier et al., 2009; Wang et al., 2018; Wang et al., 2019) for which primary emissions could play an important role."

Line 789. The results should be interpreted in terms of processes and sources.

**Reply:** We have made the following changes in the revised manuscript (page 39) to interpret the observation: "the overall hygroscopicity, and especially hygroscopicity of 150-350 particles, was highest in summer and lowest in winter at the PKU site (Beijing); one possible reason was that aerosol particles examined by Wang et al. (2018c) were most aged in the summer (and thus contained largest fractions of secondary species with high hygroscopicity) and least aged in the winter."

Line 796. That was already stated numerous times, no need to repeat. The paragraph should start with underlying reasons.

**Reply:** Indeed this has been stated many times elsewhere in the manuscript. Nevertheless, we feel that it is necessary to summarize diurnal variations reported in previous studies in the summary section, especially different diurnal variations have been reported. In our original manuscript, we have discussed underlying reasons for some of the observed diurnal variations. In the revised manuscript (page 39-40) we have expanded these sentences to provide further explanation: "The underlying reason was that photochemical processes during the daytime led to increased relative contribution of secondary aerosols, which were very hygroscopic. However, there are also exceptions. For example, $\kappa$ was larger in the nighttime than the daytime for the accumulation mode at the NKU site (Tianjin) in March 2017 (Ding et al., 2019), as high RH in the nighttime may enhance sulfate and nitrate formation from aqueous oxidation of $SO_2$ and heterogeneous hydrolysis of $N_2O_5$ (Wang et al., 2017a)."

Line 815. If kappa is considered a robust method, it does not matter at which RH GF was measured at, because lower RH would result in lower GF and kappa should be the same. If it was not the same, then that should be highlighted by proper comparison and stated clearly, because that is very important. Not all species exhibit hysteresis and even fewer when internally mixed.

**Reply:** In general we agree with the referee's concern. As a matter of fact, we compared $\kappa$ values derived from GF at different RH, and found that they were not the same. Therefore, we stated clearly. As a result, we have made the following statement clearly in our original manuscript (page 40): "Therefore, it can be concluded that using a constant $\kappa$ to describe aerosol hygroscopic growth at different RH may not always be proper.'

Line 824. NaCl has the highest deliquescence of 75% among the relevant atmospheric species, so the statement should state that no kappa (HTDMA) should be derived below 75-80%. The following Figure is manifesting that, but needs error bars added to data points.

**Reply:** In the revised manuscript (page 41) we have added error bars in Figure 5a. As the uncertainties for the data shown in Figure 5b were not provided in the original paper, we are not able to include error bars in Figure 5b, and we have added one sentence in the figure caption to explain how error bars are not displayed in Figure 5b.

We agree with the referee that no $\kappa$ should be derived from H-TDMA measurements carried out at RH before 75-80%. In Section 5 for the revised manuscript (page 53) we have made the following change to make this statement in specific: "Therefore, measurements of aerosol hygroscopicity at different RH are certainly warranted, and hygroscopic growth factors measured at high RH (at 90% RH or above) are preferably used to calculate $\kappa$ values."

Figure 5. Uncertainty of the calculated kappa is clearly above 10% based on very basic considerations. If one considers size uncertainty of two independent DMA at 10% each and RH measurement which is inherently drifting during HTDMA operation, one would get ~17% total uncertainty. Therefore, no one can objectively claim kappa differences of ~10%, because those will be within the overlapping error bars.

**Reply:** In the revised manuscript (page 41) we have included error bars in Figure 5a. We also agree with the referee's comments on uncertainties. The uncertainties shown in Figure 5a have two sources: 1) the uncertainties related to individual measurements; 2) the variation of different measurements, as only the average values from different measurements carried out at a given RH were report. Therefore, without getting access to and analyzing original data, an absolutely solid conclusion cannot be reached.

Line 877. was lower, not became lower. There is more to it. Calculated (chemical) kappa is relying on compound specific kappa values, which have uncertainty and without even mentioning rather arbitrary kappa of organic matter.

**Reply:** In the revised manuscript (page 44), we have changed "became" to "was", also added one sentence to mention the uncertainties in calculating $\kappa$ values. After revision, the last two sentences in this paragraphs have become to "This was because hygroscopicity estimated using ASCM-measured composition did not consider the contribution of smaller and less-hygroscopic particles (aerosol hygroscopicity was lower for smaller particles, but ACSM only detected >60 nm particles). In addition, the uncertainties associated with $\kappa$ values assumed for ammonium sulfate, ammonium nitrates and organics may also contribute to the discrepancies between measurement and calculation." As this is a review paper, we would like to refer readers to the original paper for further information related to $\kappa$ calculation (e.g., $\kappa$ values assumed for each individual species).

Line 883...while the increase in aerosol hygroscopicity was much smaller due to the change in chemical composition.

**Reply:** The increase in observed [CCN], was due to two reasons, i.e. increase in particle size and increased in aerosol hygroscopicity (due to change in aerosol composition). Therefore, our original statement is correct and no changes have been made.

Line 897. Was that outside uncertainty range?

**Reply:** Considering the uncertainties, some differences were very small. To provide actual $\kappa$ values (and their uncertainties) and to acknowledge the small difference, in the revised manuscript (page 45) we have made the following changes: "Compared to $\kappa$ values (increasing from 0.291±0.089 at 50 nm to 0.373±0.092 at 350 nm) derived from concurrent H-TDMA measurements, aerosol hygroscopicity derived from CCN activities were slightly lower for <50 nm particles but higher for >100 nm particles (Ma et al., 2016; Zhang et al., 2016b), but the differences were quite small."

Line 1023...and both consistencies... and discrepancies were reported

**Reply:** Corrected in the revised manuscript (page 51).

Line 1033...research directions can be proposed.

**Reply:** We have changed "discussed" to "proposed" in the revised manuscript (page 51).

Line 1042...in eastern regions

**Reply:** Corrected in the revised manuscript (page 52).

Line 1046...hygroscopicity in the cleaner troposphere. "Pristine" can only possibly apply to remote oceanic regions or Antarctica. Not even Arctic is pristine.

**Reply:** We have changed "pristine" to "cleaner" in the revised manuscript (page 52).

Line 1069. ...can be easily activated at the lowest supersaturation due to their size.

**Reply:** In the revised manuscript (page 53) we have made the following change: "…as these particles can be easily activated at low supersaturation due to their size."

Line 1074. It should be stated that kappa (HTDMA) derivation should be limited to RH above 75-80% due to reasons discussed.

**Reply:** As suggested, in the revised manuscript (page 53) we have made the following change: "Therefore, measurements of aerosol hygroscopicity at different RH are certainly warranted, and hygroscopic growth factors measured at high RH (at 90% RH or above) are preferable used to calculate $\kappa$ values."

**Reference**

Liu, H. J., Zhao, C. S., Nekat, B., Ma, N., Wiedensohler, A., van Pinxteren, D., Spindler, G., Mueller, K., and Herrmann, H.: Aerosol hygroscopicity derived from size-segregated chemical composition and its parameterization in the North China Plain, Atmospheric Chemistry and Physics, 14, 2525-2539, 2014.

Liu, P. F., Zhao, C. S., Goebel, T., Hallbauer, E., Nowak, A., Ran, L., Xu, W. Y., Deng, Z. Z., Ma, N., Mildenberger, K., Henning, S., Stratmann, F., and Wiedensohler, A.: Hygroscopic properties of aerosol particles at high relative humidity and their diurnal variations in the North China Plain, Atmospheric Chemistry and Physics, 11, 3479-3494, 2011.

Ma, N., Zhao, C., Tao, J., Wu, Z., Kecorius, S., Wang, Z., Groess, J., Liu, H., Bian, Y., Kuang, Y., Teich, M., Spindler, G., Mueller, K., van Pinxteren, D., Herrmann, H., Hu, M., and Wiedensohler, A.: Variation of CCN activity during new particle formation events in the North China Plain, Atmospheric Chemistry and Physics, 16, 8593-8607, 2016.

Meier, J., Wehner, B., Massling, A., Birmili, W., Nowak, A., Gnauk, T., Brueggemann, E., Herrmann, H., Min, H., and Wiedensohler, A.: Hygroscopic growth of urban aerosol particles in Beijing (China) during wintertime: a comparison of three experimental methods, Atmospheric Chemistry and Physics, 9, 6865-6880, 2009.

Wang, H., Lu, K., Chen, X., Zhu, Q., Chen, Q., Guo, S., Jiang, M., Li, X., Shang, D., and Tan, Z.: High N2O5 Concentrations Observed in Urban Beijing: Implications of a Large Nitrate Formation Pathway, Environmental Science & Technology Letters, 2017a.

Wang, Y., Wu, Z., Ma, N., Wu, Y., Zeng, L., Zhao, C., and Wiedensohler, A.: Statistical analysis and parameterization of the hygroscopic growth of the sub-micrometer urban background aerosol in Beijing, Atmospheric Environment, 175, 184-191, 2018c.

Wang, Y., Li, Z., Zhang, R., Jin, X., Xu, W., Fan, X., Wu, H., Zhang, F., Sun, Y., Wang, Q., Cribb, M., and Hu, D.: Distinct Ultrafine- and Accumulation-Mode Particle Properties in Clean and Polluted Urban Environments, Geophysical Research Letters, 46, 10918-10925, 2019b.

Zhang, S. L., Ma, N., Kecorius, S., Wang, P. C., Hu, M., Wang, Z. B., Groess, J., Wu, Z. J., and Wiedensohler, A.: Mixing state of atmospheric particles over the North China Plain, Atmospheric Environment, 125, 152-164, 2016b.

---

## Referee Report (RR1)

**General Comments:**

The authors have put many efforts into reviewing aerosol hygroscopicity measurements in China, which is helpful for researchers to get a quick grasp of what has been done so far regarding this topic and may provide guidance for future research in this area. Most comments raised by the two reviewers have been addressed adequately by the authors, and the manuscript is ready for publication after the following specific comments are addressed.

**Specific Comments:**

Line 208, Please be more specific, otherwise it could be misleading. Does " $\kappa$ t describe the overall aerosol properties" mean that  $\kappa$ t describes the overall aerosol hygroscopicity? If this was meant, then this statement is not correct. Assuming MAF to be 1 will certainly influence the hydrophobic part of aerosol particles, however, measured CCN activities certainly can not reflect variations of aerosol hygroscopicity of particles larger than ~ 300 nm. Thus,  $\kappa$ t might only represent overall hygroscopicity of particles that within CCN relevant diameter ranges. Overall, this is not an accurate description and should be altered.

L258 explore -> explored

Line 286, To be more precise, Wu et al., 2016 derived a linear relationship between organic aerosol hygroscopicity and O:C, which does not mean that derived  $\kappa$  of organics depended linearly on their O:C ratios. Also, one can see from Fig.8 in Wu et al., 2016 that  $\kappa_{OA}$  did not exhibit significant a linear dependence on O:C. Please rephrase this sentence.

Line 340-341: Add references to support this clarification.

Line 361: What's the difference?

Line 466: The closure results from only one site during specific periods proves nothing. Please change to "contribution of organics to aerosol hygroscopic growth was quite limited during that campaign". For example, results of Kuang et al. (2020) show that variations of organic aerosol can dominate the diurnal variations of overall aerosol hygroscopicity due to the dominant contribution of organic aerosol to aerosol mass and strong photochemical processes during daytime, which resulted in quick daytime SOA formation. Results from Jin et al. (2020) and (Li et al., 2019) also demonstrated that organic aerosol can contribute substantially to aerosol liquid water content.

Line 473: Both CCN and HTDMA measurements are not precise down to 0.001, please change 0.364 to 0.36 and also revise similar cases throughout the manuscript and the supplement materials.

Line 716: similar issue as in comment for Line 286

L916-919, The explanation for the discrepancy between ACSM calculation and CCN or HTDMA measurements needs to be improved. ACSM measures the bulk compositions of PM2.5 or PM1, so the kappa derived from ACSM measurements using volume mixing rule should be understood as the average of kappa hygroscopicity of different diameters of PM2.5 or PM1 with aerosol volume as the weight, therefore represents the overall hygroscopicity of entire aerosol population of PM1 or PM2.5. However, the HTDMA or CCN measurements only represents aerosol hygroscopicity of specified diameter or diameter range. Thus, the closure between Kappa calculated using ACSM measurements between HTDMA measurements or CCN measurements is not physically appropriate, while their variation trends may be comparable, they should not be compared against each other in closure studies. If all measurements (including aerosol chemical compositions measurements, HTDMA measurements and organic aerosol hygroscopicity) were accurate, large discrepancies can still be expected from their comparison due to their intrinsic difference in their representations of distinct aerosol populations. Volume contributions of particles with diameter < 60 nm are generally below 3% and has almost negligible impacts on Kappa calculations based on ACSM measurements, thus the inconsistency should be dominantly determined by their diameter discrepancy.

Line 1046, Give concrete values, like "larger than (range1 versus range2).....

Line 1048, Give concrete values.

Line 1127, It might be better to include most recent results on organic aerosol hygroscopicity in the North China Plain (Kuang et al., 2020) in this part.

Line 1073, Please include Liquid-Liquid phase separation

Kuang, Y., He, Y., Xu, W., Zhao, P., Cheng, Y., Zhao, G., Tao, J., Ma, N., Su, H., Zhang, Y., Sun, J., Cheng, P., Yang, W., Zhang, S., Wu, C., Sun, Y., and Zhao, C.: Distinct diurnal variation in organic aerosol hygroscopicity and its

Jin, X., Wang, Y., Li, Z., Zhang, F., Xu, W., Sun, Y., Fan, X., Chen, G., Wu, H., Ren, J., Wang, Q., and Cribb, M.: Significant contribution of organics to aerosol liquid water content in winter in Beijing, China, Atmos. Chem. Phys., 20, 901 - 914, 10.5194/acp-20-901-2020, 2020.

relationship with oxygenated organic aerosol, Atmos. Chem. Phys., 20, 865-880, 10.5194/acp-20-865-2020, 2020. Li, X., Song, S., Zhou, W., Hao, J., Worsnop, D. R., and Jiang, J.: Interactions between aerosol organic components and liquid water content during haze episodes in Beijing, Atmos. Chem. Phys., 19, 12163-12174, 10.5194/acp-19-12163-2019, 2019.

---

## Author Response (AR2)

05 October, 2020

Dear Professor Jingkun Jiang,

Thank you very much for handling our manuscript (MS No.: acp-2020-386) submitted to Atmospheric Chemistry and Physics.

Our revised manuscript, submitted on 02 September, was reviewed by three referees. One referee suggested that the manuscript can be accepted after technical correction, and minor revision was required by the other two referees. We have adequately addressed all the comments raised, and revised our manuscript accordingly again. For more information, please refer to our revised manuscript and replies to referees.

I would like to thank you and referees for all your efforts, which have largely help us improve our manuscript.

Dr. Mingjin Tang Guangzhou Institute of Geochemistry Chinese Academy of Sciences Guangzhou 510640, China Comments by referees are in blue.

Our replies are in black.

Changes to the manuscript are highlighted in red both here and in the revised manuscript.

**Reply to referee #2**

Peng et al. have adequately responded to most comments I raised. Nonetheless, some parts need further clarification and improvement. Overall, I recommend its publication in ACP with minor revisions noted below.

**Reply:** We would like to thank ref #2 for reviewing our manuscript again and recommending it for final publication after minor revision. All the comments have been addressed in our revised manuscript, as detailed below.

Line 251 in the revised text: It still lacks the in-depth interpretation of the no obvious difference in aerosol hygroscopicity between summer and winter. The authors claimed that high mass fractions of carbonaceous materials were responsible for such no seasonal trend. More elaboration on how such high mass fractions caused the no seasonality deserves to appear here. Or, at least, potential mechanisms need to be mentioned here.

**Reply:** In response to this comment, in the revised manuscript (page 12) we have modified this sentence to make the explanation concise and clear: "In addition, no obvious difference in aerosol hygroscopicity was found between summer and winter, because mass fractions of soluble inorganic species were similar in the two seasons at each individual particle size (Massling et al., 2009)."

It seems to me that the authors directly answered the question in their rebuttal on "How can the contribution of soot and organics reduce CCN activities while the hygroscopic parameters remain high?". The authors claimed in line 1042 that if measurement sites were affected by primary emissions (less hygroscopic), CCN activities could be significantly reduced. However, at the same time, the hygroscopicity in the sites was found to be high ( $\kappa$ >0.3). How can you reconcile the discrepancy? Or did I miss something?

**Reply:** Here ref #2 may misunderstand our statement. In general, CCN activities were quite high ( $\kappa$ >0.3) in these sites. However, when these sites were significantly affected by primary emissions, CCN activities would be largely reduced, and in such cases the measured  $\kappa$  values would be significantly smaller than 0.3. The work at Backgarden, Guangzhou (Rose et al., 2010, Rose et al., 2011) gave a very good examples to illustrate the effects of primary emissions (to be more specific, biomass burning) on CCN activities (see Table S5 for more details).

Line 1090: This is a relatively minor point, but it would be better if the authors can give some examples of locations that should be examined as a representative of a clean environment compared to eastern regions.

**Reply:** As suggested, in the revised manuscript (page 52) we have given a few examples for these locations: "…measurements in areas far from by human activities (e.g., Mt. Gongga in Sichuan Province, Mt. Waliguan in Qinghai Province, and Xianggelila in Yunnan Province) will be especially important…"

**Comments by referees are in blue.**

Our replies are in black.

Changes to the manuscript are highlighted in red both here and in the revised manuscript.

**Reply to referee #4**

General Comments: The authors have put many efforts into reviewing aerosol hygroscopicity measurements in China, which is helpful for researchers to get a quick grasp of what has been done so far regarding this topic and may provide guidance for future research in this area. Most comments raised by the two reviewers have been addressed adequately by the authors, and the manuscript is ready for publication after the following specific comments are addressed.

**Reply:** We would like to thank ref #4 for reviewing our manuscript and recommending it for publication after minor revision. All the comments have been properly addressed in our revised manuscript, as detailed below.

Specific Comments: Line 208, Please be more specific, otherwise it could be misleading. Does " $\kappa_t$  describe the overall aerosol properties" mean that  $\kappa t$  describes the overall aerosol hygroscopicity? If this was meant, then this statement is not correct. Assuming MAF to be 1 will certainly influence the hydrophobic part of aerosol particles, however, measured CCN activities certainly cannot reflect variations of aerosol hygroscopicity of particles larger than ~ 300 nm. Thus,  $\kappa_t$  might only represent overall hygroscopicity of particles that within CCN relevant diameter ranges. Overall, this is not an accurate description and should be altered.

**Reply:** We agree with the referee, and in the revised manuscript (page 10) we have rephrased this sentence to be more accurate: "if it is forced to be 1 (two-parameter fit), the derived activation diameter ( $d_t$ ) and single hygroscopicity parameter ( $\kappa_t$ ) describe the overall properties of aerosol particles whose diameters did not exceed the maximum diameter scanned (Rose et al., 2010)." L258 explore -> explored

**Reply:** This has been corrected in our revised manuscript (page 12).

Line 286, To be more precise, Wu et al., 2016 derived a linear relationship between organic aerosol hygroscopicity and O:C, which does not mean that derived  $\kappa$  of organics depended linearly on their O:C ratios. Also, one can see from Fig.8 in Wu et al., 2016 that  $\kappa$ OA did not exhibit significant a linear dependence on O:C. Please rephrase this sentence.

**Reply:** As suggested, in the revised manuscript (page 13) we have rephrased this sentence to be more accurate: "The measured  $\kappa$  could be well predicted using the AMS data, and a linear relationship was found between the derived  $\kappa$  of organics and their O:C ratios (Wu et al., 2016)" Line 340-341: Add references to support this clarification.

**Reply:** The work by Wang et al. (2017d) has been cited in the revised manuscript. Line 361: What's the difference?

**Reply:** As suggested, we have added several sentences in the revised manuscript (page 17-18) to further clarify the difference: "To be more specific, the average  $\kappa$  of 40 nm particles increased in daytime during clean periods due to strong photochemical reactions, while showed a reverse pattern during polluted periods due to dominant contribution by primary emissions. For 150 nm particles, average  $\kappa$  showed similar diurnal variations for clean and polluted periods, reaching maximum values at noon."

Line 466: The closure results from only one site during specific periods proves nothing. Please change to "contribution of organics to aerosol hygroscopic growth was quite limited during that campaign". For example, results of Kuang et al. (2020) show that variations of organic aerosol can dominate the diurnal variations of overall aerosol hygroscopicity due to the dominant contribution of organic aerosol to aerosol mass and strong photochemical processes during daytime, which

resulted in quick daytime SOA formation. Results from Jin et al. (2020) and (Li et al., 2019) also demonstrated that organic aerosol can contribute substantially to aerosol liquid water content.

**Reply:** We agree with the referee. In the revised manuscript (page 22) we have modified our statement to be more accurate: "...implying that the contribution of organics to aerosol hygroscopic growth was quite limited during their campaign." Line 473: Both CCN and HTDMA measurements are not precise down to 0.001, please change

0.364 to 0.36 and also revise similar cases throughout the manuscript and the supplement materials.

**Reply:** We agree with the referee that  $\kappa$  values cannot be measured with a precision down to 0.001. Nevertheless, this is a review paper and we would like to keep our values consistent with those reported in literature.

Line 716: similar issue as in comment for Line 286

**Reply:** As suggested, in the revised manuscript (page 34) we have rephrased this sentence: "A linear relationship was found between GF and O:C ratios for aerosol organics..."

L916-919, The explanation for the discrepancy between ACSM calculation and CCN or HTDMA measurements needs to be improved. ACSM measures the bulk compositions of PM2.5 or PM1, so the kappa derived from ACSM measurements using volume mixing rule should be understood as the average of kappa hygroscopicity of different diameters of PM2.5 or PM1 with aerosol volume as the weight, therefore represents the overall hygroscopicity of entire aerosol population of PM1 or PM2.5. However, the HTDMA or CCN measurements only represents aerosol hygroscopicity of specified diameter or diameter range. Thus, the closure between Kappa calculated using ACSM measurements between HTDMA measurements or CCN measurements is not physically appropriate, while their variation trends may be comparable, they should not be compared against each other in closure studies. If all measurements (including aerosol chemical compositions measurements, HTDMA measurements and organic aerosol hygroscopicity) were accurate, large discrepancies can still be expected from their comparison due to their intrinsic difference in their representations of distinct aerosol populations. Volume contributions of particles with diameter < 60 nm are generally below 3% and has almost negligible impacts on Kappa calculations based on ACSM measurements, thus the inconsistency should be dominantly determined by their diameter discrepancy.

**Reply:** We entirely agree with the referee on this point. In response, in the Section 4.5 of the revised manuscript (page 51), we have added a few sentences to further discuss the hygroscopicity closure analysis: "In hygroscopicity closure studies (either hygroscopic growth or CCN activity), average aerosol compositions are usually used to calculate hygroscopicity, and thus the calculated hygroscopicity represents the volume-weighted hygroscopicity of the entire aerosol population; on the other hand, H-TDMA and CCN measurements only provide hygroscopicity of aerosols of specific diameters or diameter ranges. As a result, although variation trends between measured and calculated hygroscopicity may be comparable, strictly speaking direct comparison is not physically appropriate. It would be more proper to compare measured hygroscopicity with that calculated using size-resolved chemical composition, as demonstrated by a closure study carried out by a campaign in central Germany (Wu et al., 2013)."

Line 1046, Give concrete values, like "larger than (range1 versus range2)..... Line 1048, Give concrete values.

**Reply:** The two comments point to the same issue and are thus addressed together. It would be nice to provide concrete values here, as suggested by the referee. However, reported  $\kappa$  values depended on particle size, and therefore it is difficult to use a few numbers to provide concrete values. Instead, in the revised manuscript (page 49) we have expanded a sentence to refer readers

to Table S5 for these numbers: "We note that a few recent studies (Atwood et al., 2017; Zhang et al., 2017; Cai et al., 2020) also reported higher aerosol hygroscopicity, as shown in Table S5. For example, the average  $\kappa$  observed at the Xinzhou site (Zhang et al., 2017) appeared to be larger than those reported at other continental sites..."

Line 1127, It might be better to include most recent results on organic aerosol hygroscopicity in the North China Plain (Kuang et al., 2020) in this part.

**Reply:** The work by Kuang et al. (Atmos. Chem. Phys., 20, 865–880, 2020) has been cited in the revised manuscript (page 54).

Line 1073, Please include Liquid-Liquid phase separation

**Reply:** In the revised manuscript (page 51) liquid-liquid phase separation has been included: "...such as solution non-ideality of aerosol droplets, limited solubility of some components contained by aerosol particles, surface tension effects, liquid-liquid phase separation, and etc."